# Cardiac differentiation of human pluripotent stem cells using defined extracellular matrix proteins reveals essential role of fibronectin

Jianhua Zhang[1,2], Zachery R Gregorich[1], Ran Tao[1], Gina C Kim[1], Pratik A Lalit[1], Juliana L Carvalho[1,3], Yogananda Markandeya[1], Deane F Mosher[1,4,5], Sean P Palecek[2,6], Timothy J Kamp[1,2,7]*

[1]Department of Medicine, School of Medicine and Public Health, University of Wisconsin - Madison, Madison, United States; [2]Stem Cell and Regenerative Medicine Center, University of Wisconsin - Madison, Madison, United States; [3]Department of Genomic Sciences and Biotechnology, University of Brasília, Brasília, Brazil; [4]Morgridge Institute for Research, Madison, United States; [5]Department of Biomolecular Chemistry, School of Medicine and Public Health, University of Wisconsin-Madison, Madison, United States; [6]Department of Chemical and Biological Engineering, College of Engineering, University of Wisconsin, Madison, United States; [7]Department of Cell and Regenerative Biology, School of Medicine and Public Health, University of Wisconsin - Madison, Madison, United States

**Abstract** Research and therapeutic applications using human pluripotent stem cell-derived cardiomyocytes (hPSC-CMs) require robust differentiation strategies. Efforts to improve hPSC-CM differentiation have largely overlooked the role of extracellular matrix (ECM). The present study investigates the ability of defined ECM proteins to promote hPSC cardiac differentiation. Fibronectin (FN), laminin-111, and laminin-521 enabled hPSCs to attach and expand. However, only addition of FN promoted cardiac differentiation in response to growth factors Activin A, BMP4, and bFGF in contrast to the inhibition produced by laminin-111 or laminin-521. hPSCs in culture produced endogenous FN which accumulated in the ECM to a critical level necessary for effective cardiac differentiation. Inducible shRNA knockdown of FN prevented Brachyury+ mesoderm formation and subsequent hPSC-CM generation. Antibodies blocking FN binding integrins α4β1 or αVβ1, but not α5β1, inhibited cardiac differentiation. Furthermore, inhibition of integrin-linked kinase led to a decrease in phosphorylated AKT, which was associated with increased apoptosis and inhibition of cardiac differentiation. These results provide new insights into defined matrices for culture of hPSCs that enable production of FN-enriched ECM which is essential for mesoderm formation and efficient cardiac differentiation.

*For correspondence:
tjk@medicine.wisc.edu

## Editor's evaluation

We found this study important for advancing derivation of cardiac cells from human pluripotent stem cells, as it convincingly supports the critical role of fibronectin in the formation of precardiac mesoderm. We believe that the work will be of interest to developmental biologists, stem cell biologists, and engineers as they work to optimize substrates used for preparation of cardiomyocytes and supporting cardiac cells.

## Introduction

Cardiomyocytes derived from human pluripotent stem cells (hPSC-CMs) are increasingly used in basic research, drug development, toxicity testing, precision medicine applications, and emerging clinical strategies for cardiac repair and regeneration. Methods to differentiate hPSC-CMs have advanced significantly over the past 15 years (*Burridge et al., 2014*; *Kattman et al., 2011*; *Lian et al., 2012*; *Mummery et al., 2012*; *Zhang et al., 2012*). Most cardiac differentiation protocols have focused on the optimal application of soluble molecules including growth factors and small molecules to promote generation of stage-specific cardiac progenitors and ultimately hPSC-CMs. These protocols also require extracellular matrix (ECM) proteins, either endogenously produced or exogenously added as substrates as well as signaling molecules to enable hPSC attachment, survival, proliferation and differentiation. However, the ECM proteins involved in the cardiac differentiation of hPSCs and ECM-activated signaling pathways have been far less investigated and elucidated.

Our previous study showed that hPSCs cultured on the commercially available ECM preparation, Matrigel, more efficiently and reproducibly differentiate to hPSC-CMs in response to Activin A/BMP4/bFGF signaling if they concurrently received overlays of Matrigel during the initiation of differentiation – the matrix sandwich protocol (*Zhang et al., 2012*). The Matrigel overlays promote the initial stage of differentiation, the epithelial-to-mesenchymal transition (EMT) to form Brachyury$^+$ mesoderm, mimicking the primitive streak in development (*Nieto et al., 2016*). However, Matrigel is a complex mixture of ECM proteins produced from Engelbreth-Holm-Swarm mouse sarcoma cells, is not fully defined, and exhibits batch-to-batch variability. The essential ECM components responsible for promoting the initial stages of cardiogenesis in the matrix sandwich protocol as well as the optimal ECM environment to promote cardiogenesis in general remain to be determined.

Complex mixtures of ECM proteins such as Matrigel allow for the attachment and self-renewal of hPSCs in appropriate media. More recently, recombinant ECM proteins and synthetic substrates have been identified that can support long-term culture of hPSCs (*Lambshead et al., 2013*). These defined substrates mimic the ECM components present in the earliest embryo including laminins, collagens, fibronectin (FN), vitronectin, and proteoglycans. The hPSCs interact with the substrates via transmembrane receptors called integrins and other cell adhesion molecules, such as cadherins. However, for cardiac differentiation protocols a substrate that both allows attachment of the hPSCs and also supports proliferation and subsequent differentiation is needed. Strong signals to maintain self-renewal and pluripotency provided by the ECM will impede the differentiation processes, so a composition of ECM that is dynamic and supports hPSC proliferation, as well as differentiation, is theoretically optimal. Yap and colleagues utilized a combination of recombinant laminins, laminin-521 (LN521) to enable self-renewal of hPSCs (*Rodin et al., 2014*) and laminin-221 (LN221) to enable differentiation to cardiac progenitors (*Yap et al., 2019*). Others using a design of experiment statistical approach found a combination of three ECM proteins optimal for cardiac differentiation of hPSCs, collagen type I, laminin-111 (LN111) and FN (*Jung et al., 2015*; *Kupfer et al., 2020*). Burridge and colleagues systematically tested a range of different substrates in a defined small molecule-based cardiac differentiation protocol and found a variety of substrates including a synthetic vitronectin-derived peptide, recombinant E-cadherin, recombinant human vitronectin, recombinant human LN521, truncated LN511 and human FN, and a FN mimetic enabled hPSC-CM differentiation (*Burridge et al., 2014*). However, these studies have not examined the impact of dynamic manipulation of defined ECM proteins or ECM signaling pathways in cardiac differentiation, nor characterized the changes in endogenous ECM proteins that ultimately contribute to the cellular transitions. In the present study, we tested a variety of human recombinant and defined ECM proteins for both attachment and overlay of hPSC cultures in the matrix sandwich protocol and investigated the potential ECM-activated signaling pathways. We chose to test LN111 (the dominant laminin isoform in Matrigel), LN521 (demonstrated adherence and culture of hPSCs [*Rodin et al., 2014*]), FN (implicated in embryonic developmental studies [*Cheng et al., 2013*; *George et al., 1993*; *Trinh and Stainier, 2004*]), and collagen (common ECM protein used for in vitro cell adherence). We found that of the tested ECM proteins, only FN overlays promoted cardiac differentiation comparable to Matrigel overlays while LN111, LN521 and collagen IV (COL4) overlays inhibited cardiogenesis. Furthermore, hPSCs differentiated efficiently to hPSC-CMs without overlay when grown on FN, LN111, and LN521. Regardless of the ECM preparation used as the attachment substrate, we identified an essential role of FN in promoting the initial stages of hPSC cardiac differentiation acting

via transmembrane integrin α4β1 and αVβ1 receptors to activate downstream integrin-linked kinase (ILK) signaling cascades.

## Results

### Defined ECM proteins support hPSC adhesion, growth, and cardiac differentiation

We tested whether defined human ECM proteins could replace Matrigel in the matrix sandwich cardiac differentiation protocol. The matrix sandwich protocol uses Matrigel coating for hPSC adhesion and expansion followed by overlaying the proliferating hPSCs with Matrigel at day –2 and day 0 of differentiation with the addition of growth factors Activin A at day 0–1, followed by addition of BMP4 and bFGF at day 1–5 (*Figure 1A*). Therefore, we first tested the ability of defined ECM proteins to support hPSC adhesion and expansion. DF19-19-11T iPSCs or H1 ESCs were seeded on human LN111, LN521, COL4, and FN coated surfaces and cultured in mTeSR1 medium. The hPSCs grew as a monolayer on LN111, LN521, and FN and exhibited similar morphology and expression of the pluripotency markers OCT4 and SSEA4 as hPSCs grown on Matrigel (*Figure 1B*, *Figure 1—figure supplement 1*). However, the hPSCs seeded on COL4 did not grow as a confluent monolayer (*Figure 1*, *Figure 1—figure supplement 2*), so this matrix coating was not tested further.

For those matrix substrates that supported monolayer growth of hPSCs, including LN111, LN521 and FN, we tested overlay of defined ECM proteins in the matrix sandwich protocol using DF19-9-11T iPSCs and H1 ESCs. Cardiac differentiation was measured by flow cytometry for cTnT$^+$ cells at 15 days of differentiation (*Figure 1C*, *Figure 1—figure supplement 3*). hPSCs seeded on Matrigel showed poor cardiac differentiation in response to the growth factors without matrix overlay, but with Matrigel overlays the percentage of cTnT$^+$ cells was significantly increased as we previously reported (*Zhang et al., 2012*). Interestingly, FN overlays were as effective as Matrigel overlays in promoting cardiac differentiation of the hPSCs growing on Matrigel. However, if cells were seeded on LN111, LN521, or FN coated surfaces, the overlay of Matrigel or FN did not further increase the efficiency of hPSC-CM generation, and overlays of LN111, LN521, and COL4 strongly inhibited cardiac differentiation. These results demonstrated that the defined ECM proteins of LN111, FN and to a lesser extent LN521 support hPSC adhesion, growth and cardiac differentiation in monolayer hPSC culture and do not require a matrix overlay for efficient cardiac differentiation using the Activin A/BMP4/bFGF growth-factor-directed protocol.

### hPSC monolayer culture on LN111 substrate promote endogenous FN production

Since FN promoted cardiac differentiation as both a culture matrix and an overlay, and also because FN plays key roles in EMT during gastrulation and cardiogenesis (*Boucaut et al., 1996*; *Boucaut and Darribere, 1983*; *Boucaut et al., 1984*; *Darribère et al., 1988*; *Johnson et al., 1993*; *Lee et al., 1984*; *Lim and Thiery, 2012*; *Linask and Lash, 1986*; *Linask and Lash, 1988a*, *Linask and Lash, 1988b*; *Nieto et al., 2016*; *Suzuki et al., 1995*; *Thiery and Sleeman, 2006*), we next examined for the presence of FN ECM in hPSC culture. DF19-9-11T iPSCs plated on Matrigel-coated surfaces and cultured in mTeSR1 per the matrix sandwich protocol were immunolabeled with a FN antibody on days –3,–2, –1 and 0 without permeabilizing the cells to examine extracellular FN protein (*Figure 2A*). On days –3 and –2, minimal immunolabeled FN in ECM was observed. However, after 4 days of culture (day 0) immunolabeled fibrillar FN ECM was abundant in the matrix sandwich culture (*Figure 2A and B*). In contrast, the monolayer culture control in the absence of Matrigel overlay had significantly less FN ECM present by day 0 (*Figure 2A and B*). This suggests that the Matrigel overlay promotes the production of endogenous FN or remodeling of FN ECM relative to the monolayer culture control without Matrigel overlay. Because the hPSCs cultured on LN111 coated surface without matrix overlay enabled efficient cardiac differentiation (*Figure 1C*, *Figure 1—figure supplement 3*), we examined the endogenous FN production in the hPSC culture on LN111 coated surface in which no exogenous FN was added. The hPSCs grown on LN111 coated surface without any matrix overlay were immunolabeled with the FN antibody without permeabilizing the cells. Confocal z-scan of the cell culture showed no detectable FN ECM at days –3 and –2, similar to the Matrigel/Matrigel sandwich culture; however, by day 0, dense fibrillar FN ECM was present in the cell culture on LN111 coated surface

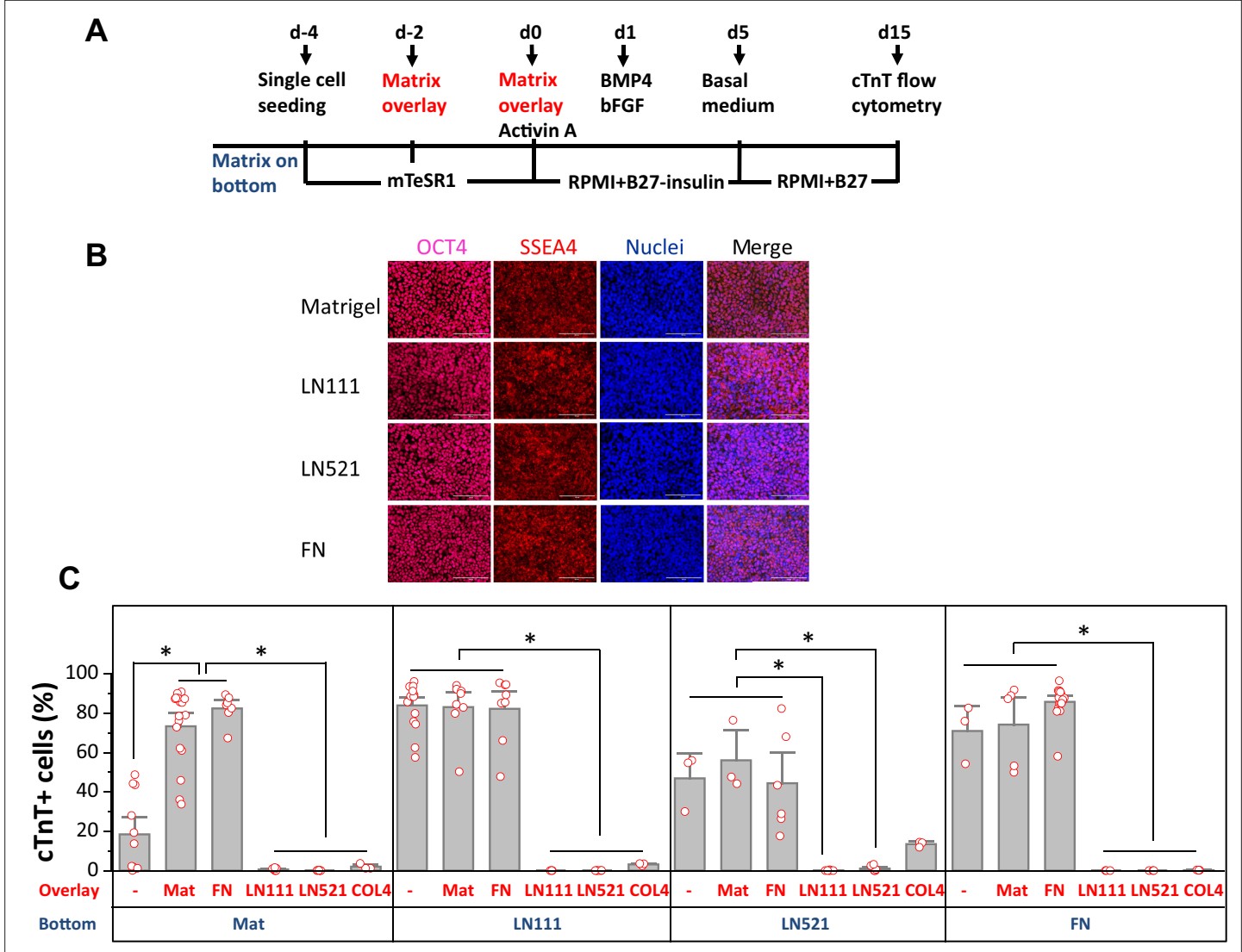

**Figure 1.** Defined ECM proteins support hPSC adhesion, growth and cardiac differentiation using the matrix sandwich protocol. (**A**) Schematic method of the matrix sandwich protocol. Defined ECM proteins were tested as coating matrix (blue) and overlay matrix (red). (**B**) Fluorescence images of DF19-9-11T iPSCs growing on the ECM of Matrigel, LN111, LN521 and FN as confluent monolayer and immuno-labeled with antibodies against OCT4 and SSEA4. Scale bar is 100 μm. (**C**) cTnT+ cells measured by flow cytometry at 15 days of differentiation of DF19-9-11T iPSCs on different ECM proteins as substrate (bottom) and overlay (overlay). N≥3 biological replicates. Error bars represent SEM. *p<0.05, one-way ANOVA with post-hoc Bonferroni test.

The online version of this article includes the following figure supplement(s) for figure 1:

**Figure supplement 1.** Fluorescence images of H1 ESCs growing on Matrigel, LN111, LN521 and FN coated surface as confluent monolayer immunolabeled with antibodies against OCT4 and SSEA4.

**Figure supplement 2.** Morphology of DF19-9-11 iPSCs growing on human collagen IV coated surface.

**Figure supplement 3.** cTnT+ cells measured by flow cytometry at 15 days of differentiation of H1 ESCs on different ECM proteins as substrate (bottom) and overlay (overlay) using the matrix sandwich protocol.

(*Figure 2C*, DF19-9-11T iPSCs; *Figure 2—figure supplement 1*, H1 ESCs). Similar to our previous study of the Matrigel/Matrigel sandwich culture (*Zhang et al., 2012*), the hPSCs growing on the LN111 coated surface without any matrix overlay formed multilayer cultures as shown in the side view of the confocal z-scan as did FN/FN matrix sandwich culture (*Figure 2D*). To determine if the hPSC culture results in accumulation of endogenously produced laminin ECM as well, DF19-9-11T iPSCs cultured on FN, LN111, and LN521 coated surfaces were immunolabeled with an antibody detecting laminins without permeabilizing the cells, and did not show measurable laminin ECM after 4 days of

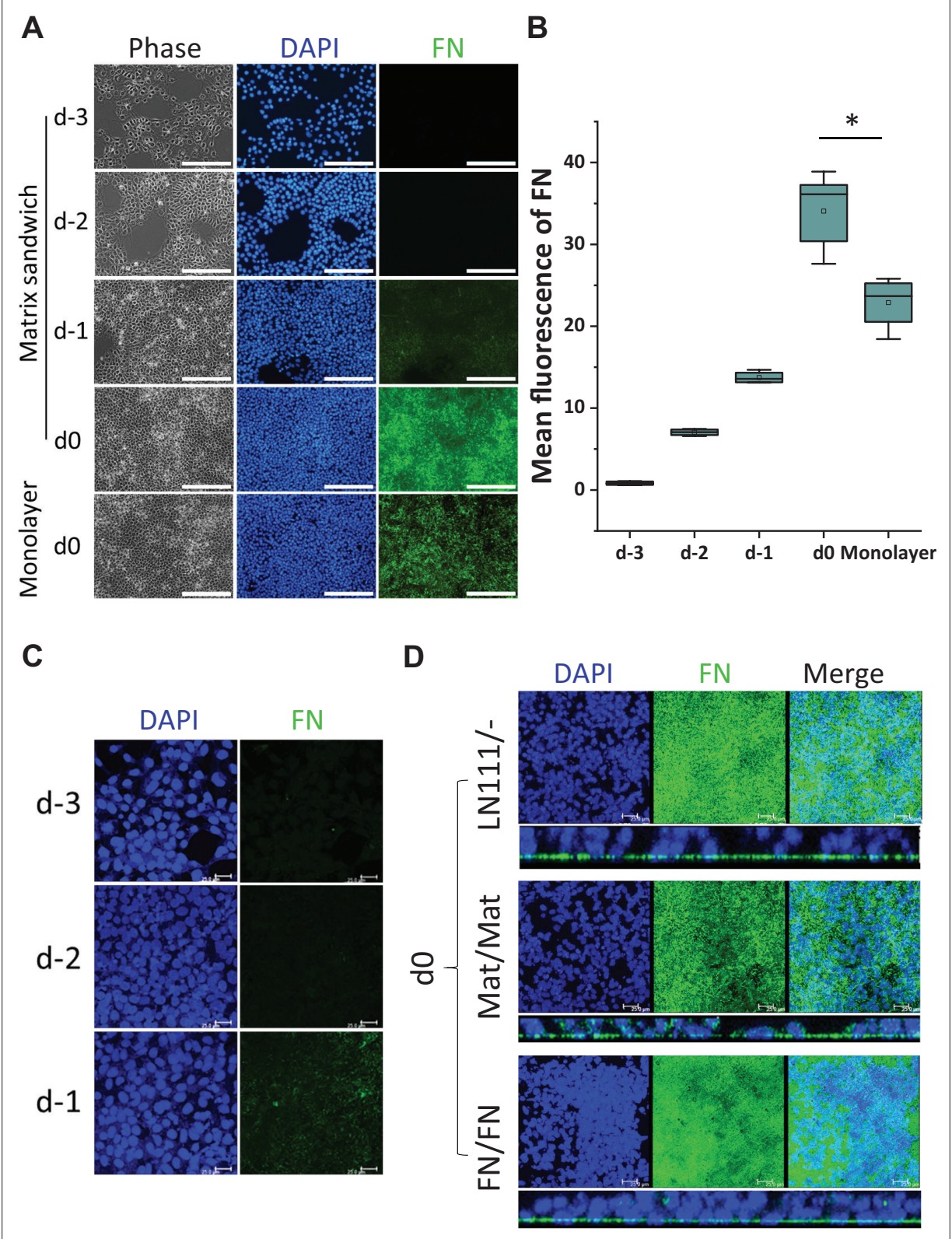

**Figure 2.** Production of endogenous FN in the hPSC-matrix sandwich culture and LN111 culture. (**A**) Phase contrast and fluorescence images of the matrix sandwich culture of DF19-9-11T iPSCs grown for 4 days and immunolabeled using anti-FN antibody, compared with the monolayer culture. Scale bar is 200 µm. (**B**) Quantitative analysis of the FN fluorescence in A by Image J. N≥3 replicates. The box plots summarize the biological replicates with the box enclosing from first to third quartile and middle square indicating mean and line in box indicating median. *p<0.05, one-way ANOVA with post-

*Figure 2 continued on next page*

*Figure 2 continued*

hoc Bonferroni test. (**C**) Maximum projection view of the confocal z-scan of DF19-9-11T iPSCs growing on LN111 coated surface immunolabeled with FN antibody at day -3, -2 and -1. Scale bar is 25 µm. (**D**) The maximum projection view (upper panel) and side view (lower panel) of the confocal z-scan of DF19-9-11T iPSCs grown for 4 days (at day 0) on LN111 coated surface without matrix overlay immunolabeled with the anti-FN antibody. The multilayer growth and FN production are similar to Matrigel/Matrigel and FN/FN matrix sandwich cultures in parallel at the same time (day 0). Scale bar is 25 µm.

The online version of this article includes the following source data and figure supplement(s) for figure 2:

**Source data 1.** Images and quantitative analysis for *Figure 2B*.

**Figure supplement 1.** Fluorescence images of H1 ESCs growing on FN, LN111 and LN521 coated surface immunolabeled with antibodies against FN and laminin.

growth on these defined matrices (*Figure 2—figure supplement 1*). These results together with the above cardiac differentiation results supported the potential role of FN in promoting ActivinA/BMP4/bFGF-directed hPSC cardiac differentiation.

## Differentiation of hPSCs on LN111 substrate undergo EMT and generate mesoderm in the FN-rich ECM

To characterize the early stages of cardiac differentiation of hPSCs cultured on LN111 and treated with Activin A/BMP4/bFGF growth factors, we examined markers of EMT, mesoderm and cardiac mesoderm. Gene expression was assessed by quantitative RT-PCR, and upon the addition of Activin A, BMP4 and bFGF, there was significant upregulation of transcription factors associated with EMT including *SNAI1* (*Leptin and Grunewald, 1990*), *SNAI2* (*Nieto et al., 1994*), and *TWIST* (*Thiery et al., 2009*; *Figure 3A*). The mesenchymal cell markers of vimentin (*VIM*), FN (*FN1*) and N-cadherin (*CDH2*) were also greatly upregulated by day 3 (*Figure 3A*). In contrast, E-cadherin expression (*CDH1*), an epithelial cadherin, was greatly downregulated by day 3 of differentiation. The mesendoderm/mesoderm transcription factors *GSC*, *MIXL1*, *SOX17* and *TBXT* were transiently upregulated followed by expression of cardiac transcription factors of *MESP1*, *ISL1*, *NKX2-5,* and *GATA4* at days 3–5 (*Figure 3B*).

To determine if FN ECM persisted or remodeled during the early stages of cardiac differentiation on LN111 substrate, immunolabeling of the early differentiated DF19-9-11T iPSCs on days 0–3 for FN and Brachyury was performed. Confocal z-scan imaging showed abundant FN ECM at each day, and Brachyury⁺ cells were associated with the dense network of FN ECM (*Figure 3C*), suggesting that the Brachyury⁺ cells interact with FN. Similar Brachyury⁺ cells and the FN ECM network were also observed in the Matrigel/Matrigel and FN/FN matrix sandwich cultures (*Figure 3—figure supplement 1*). Together, these results show that hPSCs grown on LN111, undergo the early stages of cardiac differentiation with transitions to mesoderm and cardiac mesoderm progenitors occurring in an endogenously generated FN-rich ECM, similarly as in the Matrigel/Matrigel and FN/FN matrix sandwich cultures.

## FN is essential for cardiac differentiation of hPSCs

To determine if FN is essential for mesoderm formation in our protocol and to investigate the stage-specific roles of FN during cardiac differentiation of hPSCs, we generated a doxycycline (dox) inducible FN knockdown system using *FN1* shRNA (*Figure 4—figure supplement 1*). The two vectors shown in *Figure 4A* were incorporated into lentivirus and transduced into hPSCs. Clones were selected by neomycin resistance from both hESC line H1 and hiPSC line DF19-9-11T. To confirm the dox inducibility of the *FN1* shRNA in the cell lines, we first assessed dox-induced bicistronic mCherry expression (*Figure 4—figure supplement 2A*). Inducible FN knockdown was demonstrated by immunolabeling with FN antibody (*Figure 4—figure supplement 2B*) and quantitative western blot for FN expression (*Figure 4—figure supplement 2C*).

Cardiac differentiation was performed using the monolayer based protocol with the H1 inducible FN knockdown clones growing on LN111 coated surface and treated with the growth factors Activin A, BMP4 and bFGF as shown in *Figure 4B*. To probe the stage-specific effect of FN knockdown during cardiac differentiation, dox was added at different time points: day 0–1, day 1–5, and day 5–7, and cardiac differentiation was measured by flow cytometry for cTnT⁺ cells at 15 days of differentiation. Cardiac differentiation was significantly inhibited when FN was knocked down at day 0–1, whereas FN

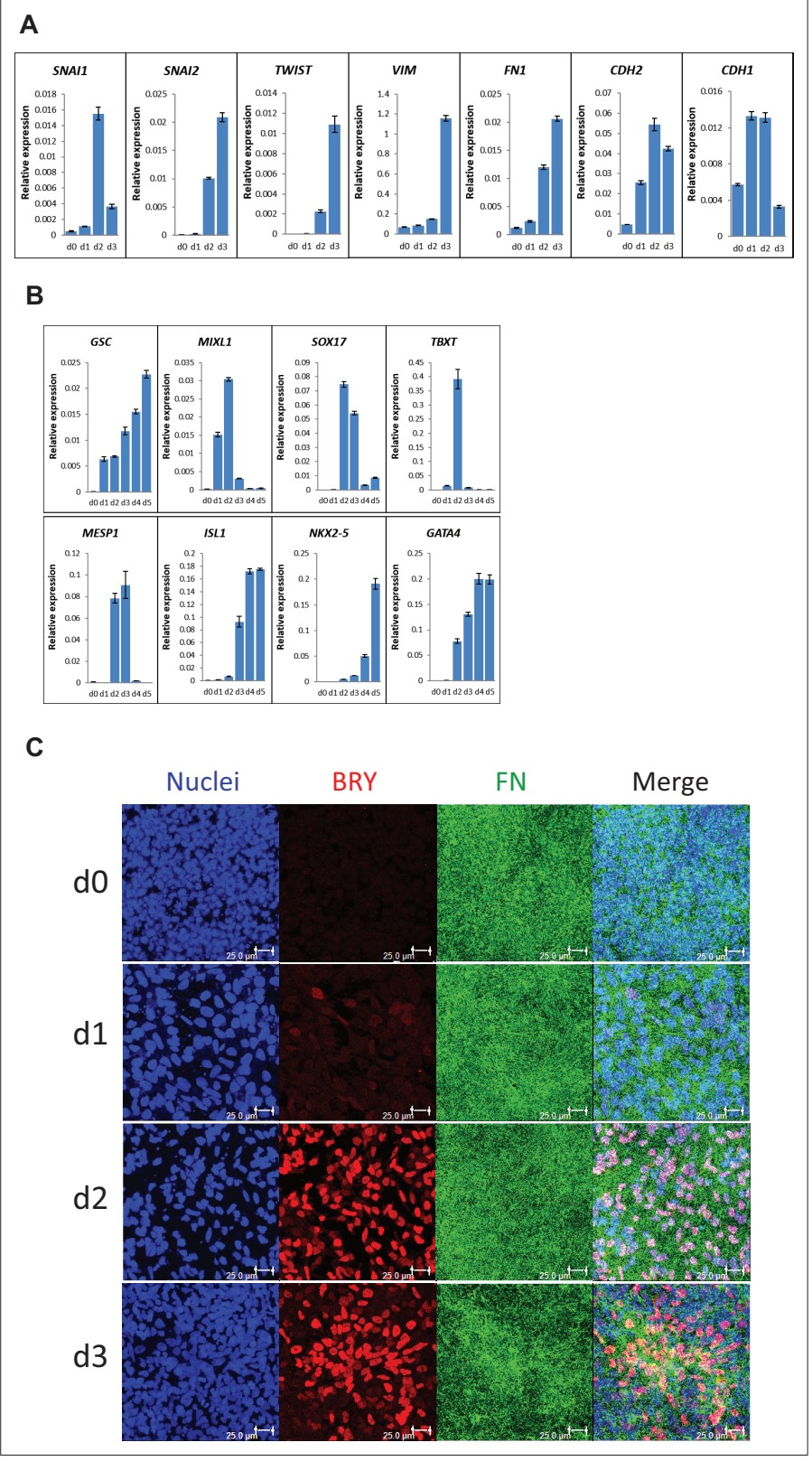

**Figure 3.** Expression of EMT, mesendoderm/mesoderm markers and cardiac transcription factors in the cardiac differentiation of hPSCs cultured on LN111 substrate by Activin A/BMP4/bFGF signaling. (**A**) qRT-PCR for gene expression of EMT markers at days 0–3 of cardiac differentiation. (**B**) qRT-PCR for gene expression of mesendoderm/mesoderm and cardiac transcription factors at days 0–5 of cardiac differentiation. N=3 technical

*Figure 3 continued on next page*

*Figure 3 continued*

replicates for each point. (**C**) Maximum projection view of the confocal z-scan of DF19-9-11T iPSCs at days 0–3 of the cardiac differentiation co-labeled with antibodies against Brachyury (BRY) and FN. Scale bar is 25 μm. Error bars represent SEM.

The online version of this article includes the following figure supplement(s) for figure 3:

**Figure supplement 1.** Maximum projection view of the confocal z-scan of DF19-9-11T iPSCs at days 0–3 of the cardiac differentiation using the Matrigel/Matrigel (Mat/Mat) and FN/FN matrix sandwich protocol.

knockdown at days 1–5, or days 5–7, did not have significant impact on the percentage of cTnT+ cells when compared to the no dox control (*Figure 4C*).

Because FN knockdown at day 0–1 significantly inhibited cardiac differentiation, we next tested if exogenous FN at this stage can rescue cardiac differentiation. Using the same protocol as shown in *Figure 4B* with dox induction of FN knockdown at day 0–1, we added soluble human FN (3 μg/cm$^2$) in the cell culture on day 0–1. Cells were differentiated for 15 days, and cardiac differentiation was measured by flow cytometry for cTnT+ cells as above. Adding exogenous FN fully rescued the hPSC-CM differentiation, giving rise to a similar percentage of cTnT+ cells compared to the no FN knockdown control (*Figure 4C*). The effect of dox-induced FN knockdown on day 0–1 was concentration-dependent (*Figure 4—figure supplement 3A*), but at the highest concentration of dox tested (8 μg/ml), there was evidence for dox toxicity based on the loss of viability of nontransgenic H1 cells undergoing the differentiation protocol (*Figure 4—figure supplement 3B*). Adding exogenous FN (3 μg/cm$^2$) along with dox at days 1–5 or days 5–7 did not significantly increase the percentage of cTnT+ cells when compared to the no dox control at days 1–5 and days 5–7, respectively (*Figure 4C*).

## FN is required for formation of Brachyury+ cells

As FN knockdown at day 0–1 dramatically inhibited cardiac differentiation which could be rescued by the addition of exogenous FN, we first evaluated the expression of EMT genes that mark the initial transition of hPSCs to mesoderm over the first two days of differentiation. Quantitative RT-PCR was performed on the same H1 inducible FN knockdown clones at 0, 24, 36, and 48 hr after hPSCs differentiation was initiated (*Figure 5A*). The effect of dox-induced knockdown of *FN1* transcripts was confirmed by the greater than 50% reduction in mRNA levels in the FN knockdown and FN knockdown +exogenous FN cell samples at 24 hr relative to the no dox control (*Figure 5A*). By 36 hr, *FN1* transcripts recovered to the control level for FN knockdown condition or were significantly increased as in the FN knockdown +exogenous FN samples after dox was removed at 24 hr. By 48 hr, there were similar levels of *FN1* transcripts in both the control and the FN rescue samples, but no cells survived in the FN knockdown group (*Figure 5A*). The key EMT transcription factors upregulated during gastrulation (*Barrallo-Gimeno and Nieto, 2005*; *Nieto, 2002*; *Nieto et al., 2016*; *Thiery and Sleeman, 2006*), *SNAI1 and SNAI2,* were examined. Quantitative RT-PCR showed *SNAI1* expression increased significantly in both the FN knockdown and FN knockdown +exogenous FN samples compared to the control at 24 hr, and its expression continuously upregulated in the FN knockdown sample at 36 hr. By 48 hr, there were similar level of *SNAI1* expression in both the control and the FN rescue samples; whereas, *SNAI2* expression was not significantly different between the groups at 24 and 36 hr but a general increase in expression over this time window was observed (*Figure 5A*). *VIM* expression, similar to *SNAI1* expression, was increased significantly in the FN knockdown cells by 36 hr compared to the control (*Figure 5A*), which is consistent with this mesenchymal marker and known target of *SNAI1*.

We next examined the mesendodermal/mesodermal progenitors generated in the initial differentiation stage of the H1 inducible FN knockdown clones by flow cytometry. The cell counts of the attached cells on days 0–5 of differentiation showed a great reduction of cell number at day 1 in all three groups (*Figure 5B*); however, the cells in the control and the FN rescue groups rapidly proliferated after day 1. In contrast, no cells survived in the FN knockdown group after day 2 (*Figure 5B*). Because Brachyury and Sox17 are both expressed in mesendodermal progenitors, we co-labeled the cells with Brachyury and Sox17 antibodies on days 0–5 and analyzed by flow cytometry. Brachyury+ cells started to emerge at day 1 in all three groups. By day 2, 97–98% of the cells were Brachyury+ in both the control and FN rescue groups, but there were no surviving cells in the FN knockdown samples (*Figure 5C*). The fraction of cells that were Brachyury+ rapidly decreased after day 2 in both

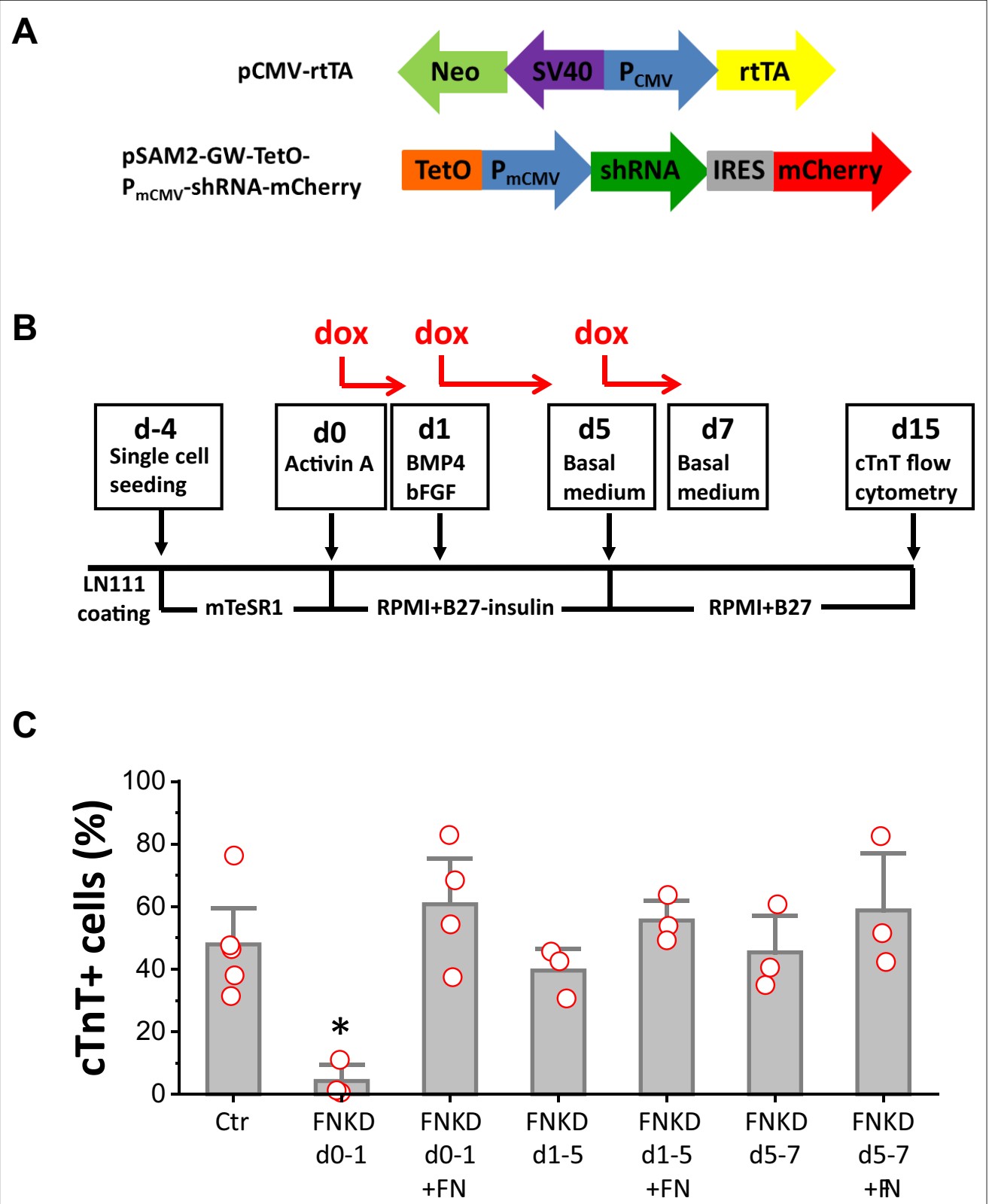

**Figure 4.** FN is essential at the initiation of cardiac differentiation of hPSCs. (**A**) Schematic of the inducible shRNA construct for *FN1* knockdown. (**B**) Schematic method of FN knockdown at differentiation stages of days 0–1, 1–5, and 5–7 in the cardiac differentiation protocol. (**C**) cTnT+ cells measured by flow cytometry at 15 days of differentiation of the H1 *FN1* knockdown clone 34 using the protocol in (**B**). Dox concentration is 2 µg/ml,

*Figure 4 continued on next page*

*Figure 4 continued*

N≥3 biological replicates. Error bars represent SEM. *p<0.05, one-way ANOVA with post-hoc Bonferroni test. FNKD indicates FN knockdown by dox induction. +FN indicates exogenous FN added.

The online version of this article includes the following figure supplement(s) for figure 4:

**Figure supplement 1.** The *FN1* shRNA oligos.

**Figure supplement 2.** Characterization of hPSC *FN1* knockdown clones.

**Figure supplement 3.** Effect of dox concentration-dependent FN knockdown at days 0–1 on cardiac differentiation using the protocol as shown in *Figure 4A*.

**Figure supplement 4.** The cloning strategy in generation of inducible *FN1* knockdown hPSC clones.

**Figure supplement 5.** The concentrations of G418 tested for neo-resistant selection of hPSC clones.

the control and FN rescue groups, and by day 5 there were minimal Brachyury+ or Sox17+ cells present in both groups (*Figure 5C*). We performed the same experiment using the DF19-9-11T inducible FN knockdown clones and observed similar results (*Figure 5—figure supplement 1*). In contrast to the day 0–1 treatment with dox, a later timed dox pulse from days 1 to 5 did not alter the abundance of Brachyury+ cells in all three groups over the same time course of differentiation (*Figure 5—figure supplement 2*). Together these results suggest that knockdown of FN at day 0–1 does not stop the initiation of EMT upon addition of Activin A at d0 (*Nieto et al., 2016*; *Thiery and Sleeman, 2006*), but it prevents the generation and/or survival of Brachyury+ mesodermal progenitors. In contrast, shRNA knockdown of FN at later time points in the protocol did not impact the fate of the differentiating cells.

## Addition of blocking antibodies to integrin β1, α4 or αV subunits at mesoderm formation inhibits hPSC-CM differentiation

To investigate the mechanisms underlying FN's essential role in the formation of Brachyury+ mesoderm, we evaluated integrins expressed in undifferentiated hPSCs and known to bind FN. Integrins are a family of heterodimeric transmembrane proteins composed of α and β subunits that interact with ECM. Of the 24 known heterodimeric integrin receptors, 13 have been shown to bind FN (*Bachmann et al., 2019*; *Bharadwaj et al., 2017*; *Hynes, 2002*; *Ruoslahti, 1991*; *Wu et al., 1995*). Review of RNA-seq data from undifferentiated hiPSCs shows expression of integrin subunits associated with FN binding including integrin α3, α4, α5, αV, β1, β5, and β8 (*Zhang et al., 2019*). Of these integrin subunits, knockout studies have implicated only α4, α5, αV, and β1 with various developmental defects impacting the heart (*Hynes, 2002*), so we focused our studies on these integrins. Integrin β1 shows the highest level of expression of all integrin subunits in hPSCs, and so we first tested blocking integrin β1 with a monoclonal antibody, P5D2, during differentiation. We added P5D2 at day –2 through day –1 (pluripotent stage) or day 0 through day 1 (mesoderm formation) in the matrix sandwich protocol when Matrigel overlays were applied (*Figure 6A*). Adding P5D2 at day –2 did not block cardiac differentiation; however, adding P5D2 at day 0 significantly inhibited cardiac differentiation as measured by flow cytometry of the cTnT+ cells using DF19-9-11T iPSCs and H1 ESCs (*Figure 6B*, *Figure 6—figure supplement 1A*). Furthermore, the P5D2 antibody showed concentration-dependent inhibition of hPSC-CM generation in DF19-9-11T iPSCs and H1 ESCs (*Figure 6C*, *Figure 6—figure supplement 1B*). We next tested antibody blocking (3 µg/ml) of relevant integrin α subunits including α5, αV, and α4. Adding the monoclonal antibodies P1D6 (anti-integrin α5), P3G8 (anti-integrin αV), or P4G9 (anti-integrin α4) at day –2 as shown in *Figure 6A* did not significantly impact hPSC-CMs differentiation as measured by flow cytometry of the cTnT+ cells, similar to the results blocking integrin β1 at day –2 (*Figure 6D*, *Figure 6—figure supplement 1C*). When the integrin α blocking antibodies were added on day 0, block of integrin α5 did not show inhibition of hPSC-CMs differentiation for both DF19-9-11T and H1 lines. Whereas block of integrin αV showed borderline inhibition and block of integrin α4 showed significant inhibition of hPSC cardiac differentiation in both DF19-9-11T and H1 lines (*Figure 6D*, *Figure 6—figure supplement 1C*). To probe the impact of blocking integrin α4 and integrin αV further, we tested a range of blocking antibody concentrations added at day 0 and found concentration-dependent inhibition of cardiac differentiation by P4G9 (anti-integrin α4), as well as significant inhibition by the highest concentration (5 µg/ml) of P3G8 (anti-integrin αV) (*Figure 6E and F*). Taken together, these results showed antibody blocking FN integrin receptors of β1, α5, αV, or

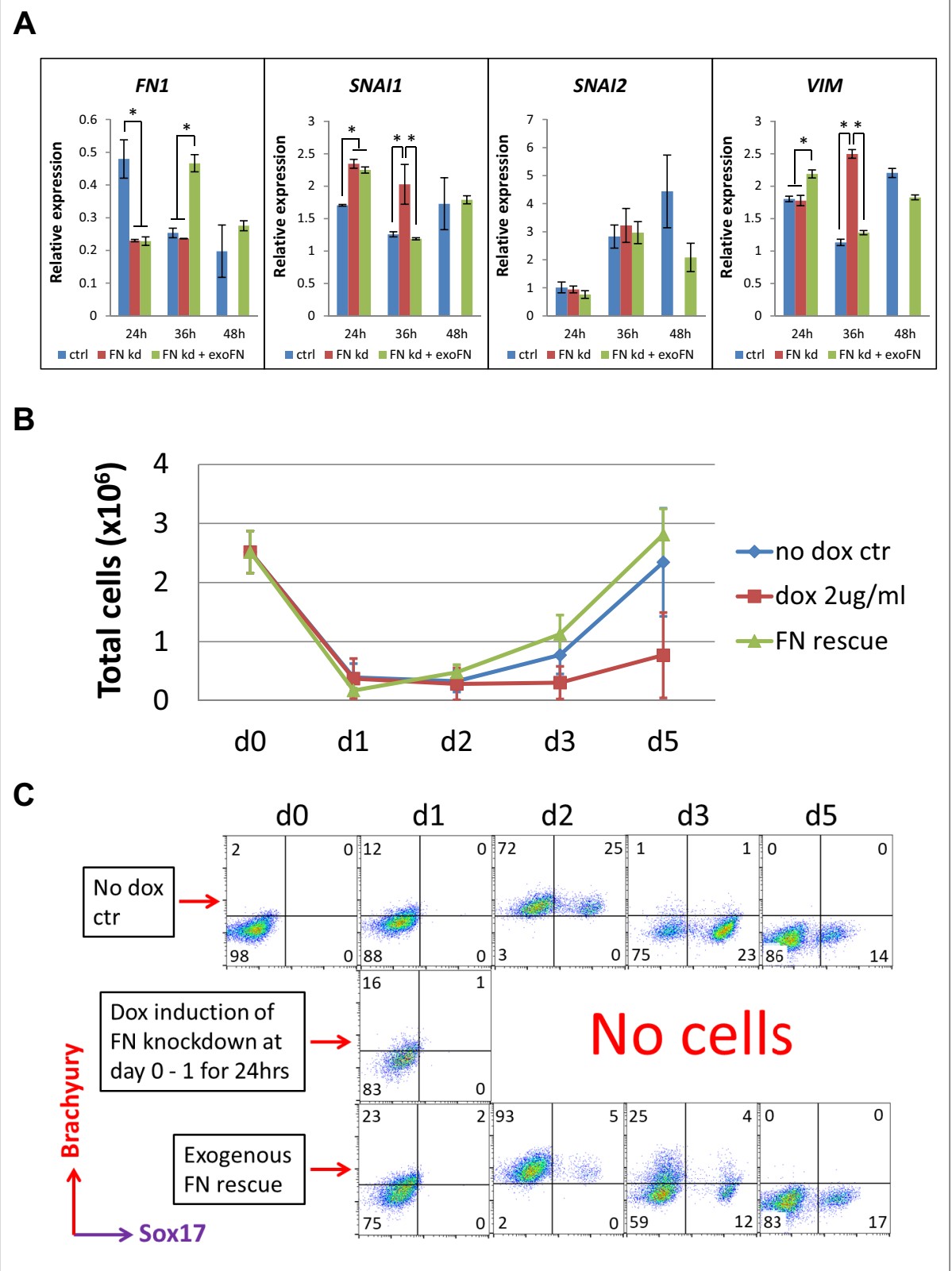

**Figure 5.** shRNA knockdown of FN results in loss of Brachyury⁺ cells. (**A**) qRT-PCR for gene expression of EMT markers for the H1 FN knockdown clone in the cardiac differentiation time course of 0–48 hr at the no dox control, dox induction at day 0–1 and dox induction at day 0–1 with adding exogenous FN conditons. (**B**) Total cell number of the H1 FN knockdown clones in the time course of days 0–5 in the cardiac differentiation at the no dox control, dox induction at day 0–1 and dox induction at day 0–1 with adding exogenous FN conditions. (**C**) Flow cytometry of co-labeling the cells shown in B with

*Figure 5 continued on next page*

*Figure 5 continued*

Brachyury and Sox17 antibodies. The cardiac differentiation protocol is shown in *Figure 4B*. Error bars represent SEM. *p<0.05, one-way ANOVA with post-hoc Bonferroni test.

The online version of this article includes the following figure supplement(s) for figure 5:

**Figure supplement 1.** Flow cytometry co-labeling cells with antibodies to Brachyury and Sox17 using 19-9-11 FN knockdown clone 42 in the cardiac differentiation protocol at days 0–5, comparing no dox control, dox induction at day 0–1 and dox induction at day 0–1 with addition of exogenous FN conditions.

**Figure supplement 2.** Flow cytometry co-labeling cells with antibodies to Brachyury and Sox17 using H1 FN knockdown clone 8 in the cardiac differentiation protocol at days 0–5, comparing no dox control, dox induction at days 1–5 and dox induction at days 1–5 with addition of exogenous FN conditions.

α4 at the pluripotent stage of hPSCs (day –2) in the matrix sandwich protocol did not inhibit cardiac differentiation; however, blocking integrin β1, α4, or αV at day 0–1, when mesodermal progenitors are generated resulted in significant inhibition of hPSC-CM differentiation compared to the control for multiple hPSC lines. Thus, integrin α4β1 and αVβ1 heterodimers are likely key mediators of the FN effect on early differentiation stages.

## Integrin-linked kinase signaling promotes cardiac differentiation

Integrin-linked kinase (ILK) interacts with the cytoplasmic domain of β1 and β3 integrins and plays crucial roles in transducing signals from ECM components and growth factors to downstream signaling components (*Delcommenne et al., 1998*; *Hannigan et al., 1996*; *Oloumi et al., 2004*). The kinase activity of ILK is stimulated rapidly and transiently by the engagement of integrins to FN ECM, as well as by insulin via receptor tyrosine kinases (RTKs), in a PI3K dependent manner (*Delcommenne et al., 1998*). To determine if ILK signaling is involved in the initiation of cardiac differentiation, we tested the specific ILK inhibitor, cpd22 (*Lee et al., 2011*), in the matrix sandwich protocol. Cpd22 was added at day 0 at concentrations of 0, 1, 3, and 10 μM, and cells were treated for 24 hr in the matrix sandwich protocol as shown in *Figure 7A*. Cardiac differentiation was measured by flow cytometry of the cTnT$^+$ cells at 15 days of differentiation. Cpd22 showed significant, concentration-dependent inhibition of cardiac differentiation in both DF19-9-11T iPSCs and H1 ESCs (*Figure 7B*, *Figure 7—figure supplement 1*).

We sought to confirm the role of ILK in the initiation of cardiac differentiation in a distinct hPSC-CM differentiation protocol based on biphasic modulation of Wnt signaling by small molecules (*Lian et al., 2012*; *Lian et al., 2013*). This protocol involves the activation of canonical Wnt signaling by inhibition of GSK3β to promote Brachyury$^+$ mesoderm formation, followed by inhibition of Wnt to promote hPSC-CM differentiation (GiWi protocol). Cpd22 was added on day 0 at concentrations of 0, 1, 3, and 10 μM and cells were treated for 24 hr in the GiWi protocol as shown in *Figure 7C*. Cardiac differentiation was measured by flow cytometry of the cTnT$^+$ cells at 15 days of differentiation. Cpd22 showed significant inhibition of cardiac differentiation in a concentration-dependent manner in the GiWi protocol (*Figure 7D*), similar to the matrix sandwich protocol (*Figure 7B*). Collectively these results show that ILK plays an important role in mesoderm formation and cardiac differentiation of hPSCs potentially acting downstream of integrin β1 and FN.

## Inhibition of ILK promotes apoptosis of differentiating hPSCs at mesoderm formation

Because inhibition of ILK by cpd22 significantly inhibited cardiac differentiation in the matrix sandwich protocol, we next investigated apoptosis after 18 hr of cpd22 addition on day 0 in the matrix sandwich protocol. To identify apoptotic and necrotic cells, annexin V/ propidium iodide (PI) double staining was used in flow cytometry for both DF19-9-11T iPSCs and H1 ESCs. Annexin V has a high affinity for the anionic phospholipid phosphatidylserine (PS) specifically present in the outer leaflet (extracellular side) of the plasma membrane in apoptotic cells. PI is used to detect necrotic or late apoptotic cells, characterized by the loss of the integrity of the plasma and nuclear membranes. Cells were collected for flow cytometry at 18 hr of differentiation in the matrix sandwich protocol in the absence or presence of cpd22 at 1, 3, and 10 μM added on day 0 as shown in *Figure 7A*. The flow cytometry data plots can be divided in four regions corresponding to: 1. viable cells which are Annexin

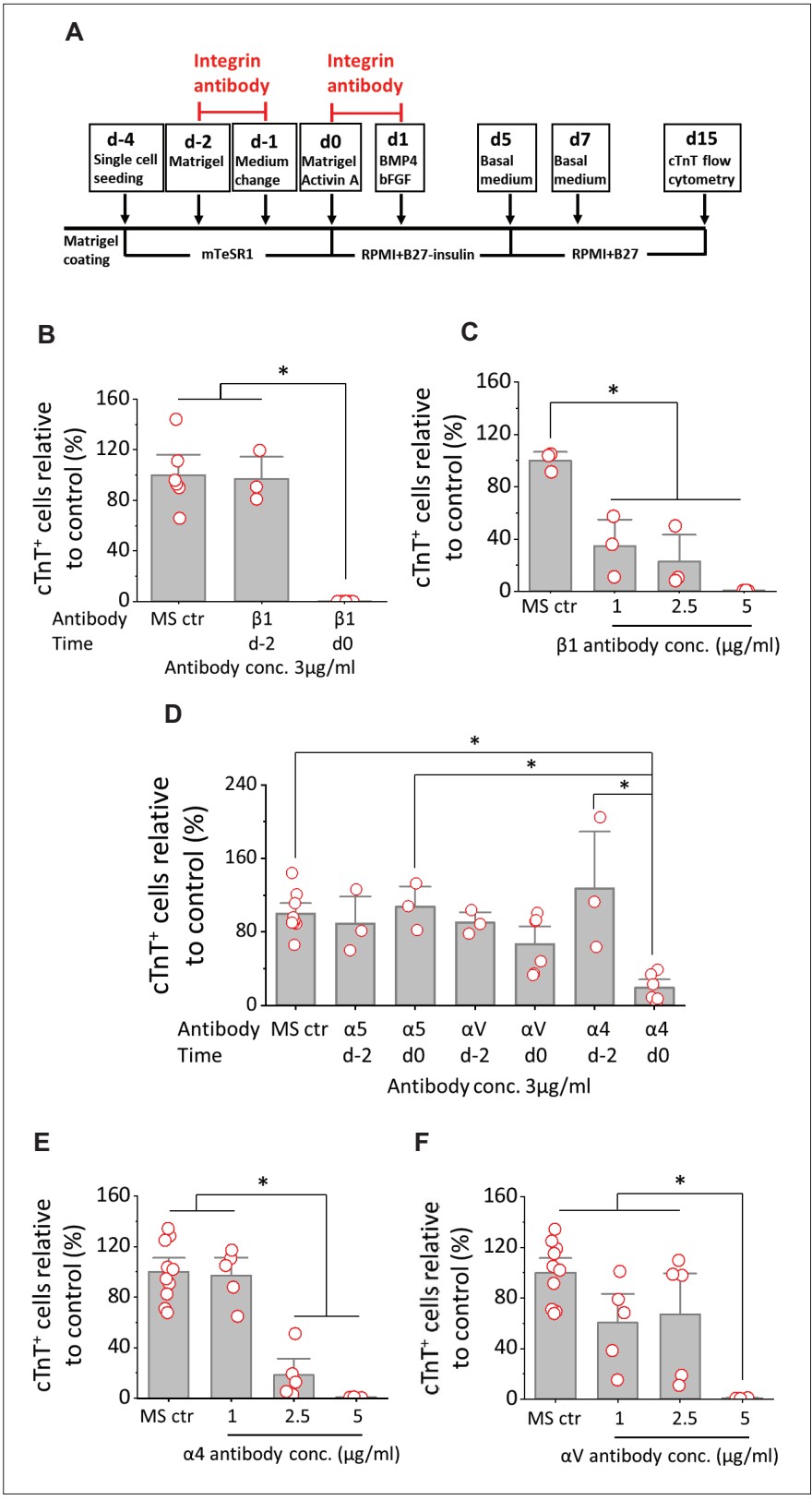

**Figure 6.** Cardiac differentiation is blocked by anti-integrin β1, α4 or αV antibodies when added at mesoderm formation in the matrix sandwich protocol. (**A**) Schematic for testing monoclonal antibodies to block integrin β1 (P5D2), α5 (P1D6), αV (P3G8), and α4 (P4G9). (**B**) cTnT+ cells measured by flow cytometry at 15 days differentiation when anti-human integrin β1 antibody was added on day –2 or day 0 at 3 μg/ml in the matrix sandwich protocol

*Figure 6 continued on next page*

*Figure 6 continued*

as shown in A. (**C**) cTnT$^+$ cells measured by flow cytometry at 15 days differentiation when anti-human integrin β1 antibody was added on day 0 at concentrations of 1, 2.5, and 5 µg/ml in the matrix sandwich protocol as shown in A. (**D**) cTnT$^+$ cells measured by flow cytometry at 15 days differentiation when anti-human integrin α5, αV or α4 antibody was added on day –2 or day 0 at 3 µg/ml in the matrix sandwich protocol as shown in A. (**E**) cTnT$^+$ cells measured by flow cytometry at 15 days differentiation when anti-human integrin α4 antibody was added on day 0 at concentrations of 1, 2.5, and 5 µg/ml in the matrix sandwich protocol. (**F**) cTnT$^+$ cells measured by flow cytometry at 15 days differentiation when anti-human integrin αV antibody was added on day 0 at concentrations of 1, 2.5, and 5 µg/ml in the matrix sandwich protocol. The % of cTnT$^+$ cells in each group was normalized to the control group to combine replicates from multiple differentiations and to compare across different antibodies blocking. N≥3 biological replicates. Data are from DF19-9-11T iPSC line. Error bars represent SEM. *p<0.05, one-way ANOVA with post-hoc Bonferroni test.

The online version of this article includes the following figure supplement(s) for figure 6:

**Figure supplement 1.** Cardiac differentiation of H1 ESCs is blocked by anti-integrin β1, α4, or αV antibody when added on day 0 in the matrix sandwich protocol.

V$^-$/PI$^-$ (Q4); 2. early apoptotic cells which are Annexin V$^+$/PI$^-$ (Q1); 3. late apoptotic cells which are Annexin V$^+$/PI$^+$ (Q2); and 4. necrotic cells which are Annexin V$^-$/PI$^+$ (Q3) (*Figure 8A*, *Figure 8—figure supplement 1A*). Under control conditions for matrix sandwich protocol, a substantial fraction of cells were undergoing apoptosis by 18 hr for both DF19-9-11T (71.6%, total Annexin V$^+$ cells, *Figure 8A*) and H1 (47.6%, *Figure 8—figure supplement 1A*) lines. This cell death is typical of monolayer-based cardiac differentiation protocols in response to cell signaling (Activin A) and media conditions that select for survival specifically of mesendodermal fated cells. However, addition of cpd22 resulted in a concentration-dependent increase in the total Annexin V$^+$ cells (*Figure 8B*, *Figure 8—figure supplement 1B*). Furthermore, 10 µM cpd22 significantly increased the population of late apoptotic cells (*Figure 8C*, *Figure 8—figure supplement 1C*).

Because cpd22 also inhibited cardiac differentiation in the GiWi protocol, we tested if inhibition of ILK by cpd22 also promoted apoptosis in the first 24 hr of the GiWi protocol. Using the same apoptosis assay, cells were collected for flow cytometry after 18 hr of differentiation in the GiWi protocol in the absence or presence of cpd22 at 1, 3, and 10 µM as shown in *Figure 7C*. Similar to the matrix sandwich protocol, the GiWi control cells exhibited significant apoptosis (58.5%, total Annexin V$^+$ cells of DF19-9-11T, *Figure 8D*) at 18 hr of differentiation. Adding cpd22 at 10 µM significantly increased the total Annexin V$^+$ cells as well as the late apoptotic cells (*Figure 8E and F*). This finding of increased apoptosis in response to cpd22 for both the matrix sandwich and GiWi protocols correlates with the inhibition of cardiac differentiation by cpd22 in these protocols, and thus suggests that survival of the critical mesodermal population fated for cardiac differentiation requires ILK signaling.

## Inhibition of ILK at mesoderm formation reduces phosphorylation of AKT without changing phosphorylation of GSK3β

To investigate how ILK regulates cells survival and differentiation, we examined regulation of AKT and GSK3β, known downstream targets of ILK signaling (*Delcommenne et al., 1998*; *Legate et al., 2006*). Activation of ILK can lead to phosphorylation of AKT on serine-473 which inhibits apoptosis and promotes cell survival. ILK activation by the ECM and/or growth factors in PI3K-dependent manner, can increase phosphorylation of GSK3β at serine-9 which inhibits GSK3β activity. Quantitative Western blots for AKT, pAKT (S473), GSK3β, and pGSK3β (S9) were performed on cells undergoing the matrix sandwich protocol in the absence or presence of the ILK inhibitor, cpd22 (10 µM), added at day 0 (*Figure 9*). The pAKT level (normalized to total protein) at 10, 30, and 120 min showed a significant decrease at 120 min by cpd22, but no significant differences at other time points (*Figure 9A and B*). At the 24 hr time point, an inadequate number of cells survived in the presence of cpd22 to provide a sample. The pGSK3β level (normalized to total protein) at the same time points in the protocol did not show any significant effect of cpd22 (*Figure 9D and E*). Treatment with cpd22 did not significantly change the levels of total AKT or total pGSK3 over the time course (*Figure 9A, C, D and F*). Together, these results are consistent with ILK inhibition reducing pAKT levels and thus leading to increased apoptosis and associated failure of cardiac differentiation.

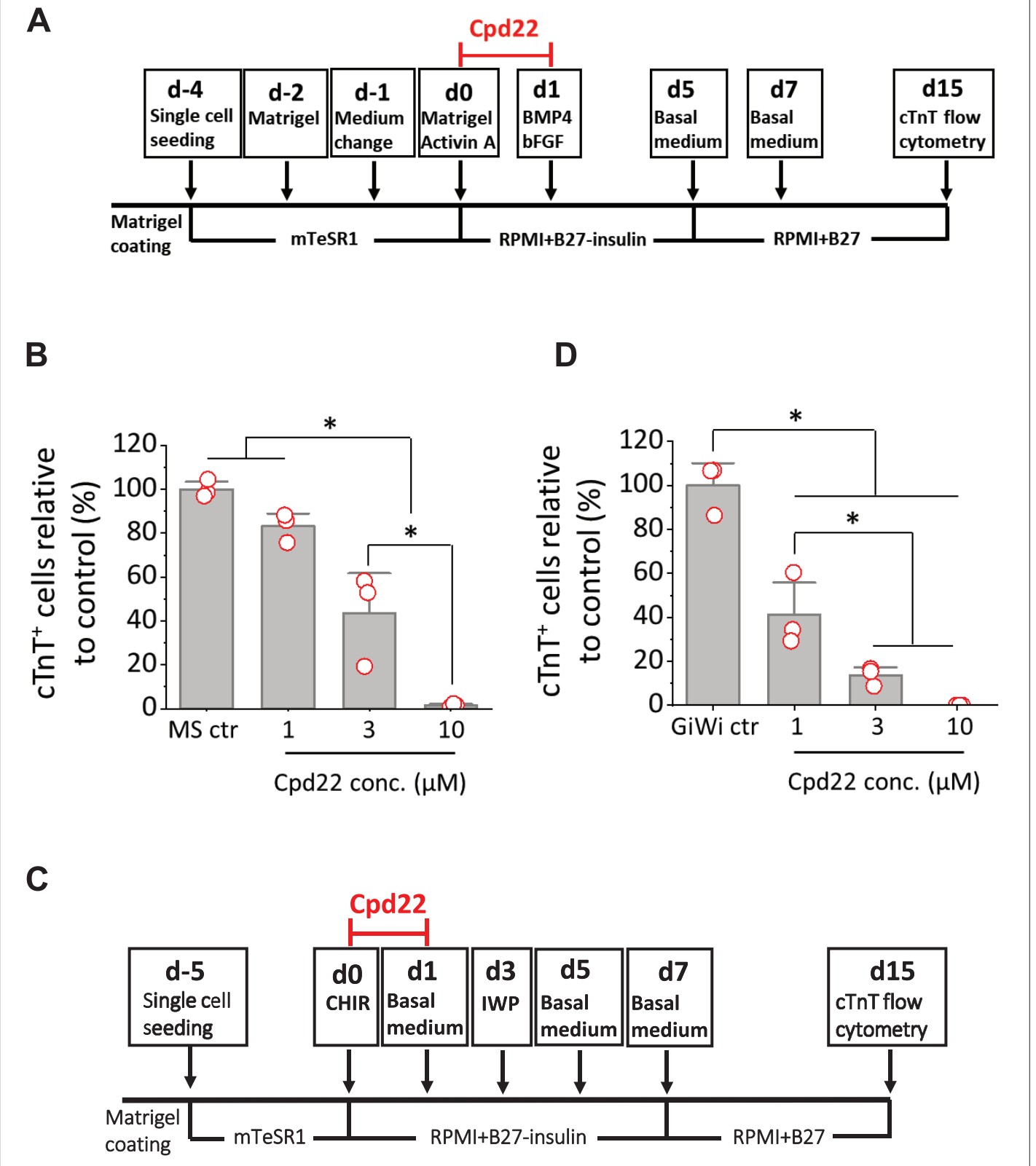

**Figure 7.** Cardiac differentiation is inhibited by ILK inhibitor in the matrix sandwich and GiWi protocols. (**A**) Schematic to test inhibition of ILK by small molecule inhibitor Cpd22 in the matrix sandwich protocol. (**B**) cTnT+ cells measured by flow cytometry at 15 days differentiation when Cpd22 was added at day 0 at concentrations of 1, 3, and 10 μM in the matrix sandwich protocol as shown in A. (**C**) Schematic for testing ILK inhibitor Cpd22 in the GiWi protocol. (**D**) cTnT+ cells measured by flow cytometry at 15 days differentiation when Cpd22 was added on day 0 in the GiWi protocol as shown in C. The

*Figure 7 continued on next page*

*Figure 7 continued*

% of cTnT$^+$ cells in each group was normalized to the control group to combine replicates in multiple differentiations and to compare across different antibodies. N≥3 biological replicates. Data are from DF19-9-11T iPSC line. Error bars represent SEM. *p<0.05, one-way ANOVA with post-hoc Bonferroni test.

The online version of this article includes the following figure supplement(s) for figure 7:

**Figure supplement 1.** Cardiac differentiation of H1 ESCs is inhibited by ILK inhibitor, cpd22 when added on day 0 at concentrations of 0, 1, 3 and 10 uM in the matrix sandwich protocol.

## Discussion

Here, we identify defined ECM proteins including LN111, LN521 and FN that can support the attachment of hPSCs and subsequent hPSC-CM differentiation in coordination with Activin A/BMP4/bFGF signaling. In addition, we demonstrate that initiating the differentiation process with an overlay of defined ECM protein can either promote (FN) or inhibit (LN111, LN521 and COL4) cardiogenesis. Regardless of seeding substrate, cardiac differentiation starts with the generation of Brachyury$^+$ cells which is dependent on adequate FN in the ECM either generated endogenously or provided exogenously. FN interacts with integrins β1, α4, and αV on the differentiating hPSCs potentially activating ILK signaling and increasing pAKT levels which contributes to the formation of Brachyury$^+$ mesoderm. These results provide new insights into defined matrices for cardiac differentiation of hPSCs and the essential role of FN in the earliest stages of cardiac differentiation.

The Activin A/BMP4/bFGF-directed hPSC cardiac differentiation protocols were initially developed using the embryoid body (EB) approach starting with suspended aggregates of hPSCs (*Dubois et al., 2011*; *Kattman et al., 2011*; *Yang et al., 2008*). Using the combination of Activin A/BMP4/bFGF at the initial differentiation stage in the protocol induces mesoendoderm formation consistent with known signaling pathways during embryonic development (*Conlon et al., 1994*; *Lough et al., 1996*; *Mima et al., 1995*; *Winnier et al., 1995*). However, this growth factor-directed cardiac differentiation exhibits variability when applied either in the EB type protocols or monolayer-based hPSC differentiation approaches (*Kattman et al., 2011*; *Laflamme et al., 2007*; *Zhang et al., 2012*). The lack of spatial organization and morphogen gradients in these in vitro differentiation approaches relative to embryonic development likely contribute to the variability. To address this variability, we developed the matrix sandwich method to promote the monolayer-based, growth factor-directed cardiac differentiation of hPSCs (*Zhang et al., 2012*). Although the matrix sandwich method promoted the EMT during the initial differentiation stage which is required for Brachyury$^+$ mesoderm formation, the mechanism was unclear due to the complex mixture of ECM proteins in Matrigel. In the present study, we demonstrate that the accumulation and assembly of endogenously produced FN ECM induced by Matrigel overlays is a critical enabling feature of the protocol to promote formation of Brachyury$^+$ mesoderm and allow for successful cardiac differentiation.

A rapid increase in FN ECM is a well-described feature of metazoan gastrulation in studies from sea urchin to mouse (*Duband and Thiery, 1982*; *George et al., 1993*; *Spiegel et al., 1980*). *Fn* null mouse embryos generate mesoderm but have profound heart developmental defects with early embryonic lethality (*George et al., 1993*). Mutant analysis in zebrafish linked cardiac bifida and impaired cardiac mesoderm migration with a mutation in *nat* which encodes FN (*Trinh and Stainier, 2004*). A critical role of FN produced by endoderm in the formation of precardiac mesoderm was suggested by a study using mouse ESCs (*Cheng et al., 2013*). The present results using hPSCs also suggest a critical role of FN in cardiogenesis but suggest that FN is required for the formation and survival of Brachyury$^+$ mesoderm, potentially an earlier developmental defect than identified in the mouse and zebrafish studies. Whether this failure to form mesoderm in the absence of FN from hPSCs is a manifestation of the particular in vitro differentiation protocol used or reflects a critical role of FN earlier in human development will require future studies.

The role of FN in hPSC cardiac differentiation is likely multifactorial. FN interacts with cells by binding integrin receptors and activating ILK signaling cascades including PI3K/AKT, growth factor/RTKs, GSK3 and canonical Wnt/β-catenin signaling pathways to regulate cell survival, proliferation, migration, and differentiation (*Delcommenne et al., 1998*; *Oloumi et al., 2004*). Our results demonstrating that antibody block of integrin β1, α4, or αV subunits mimics the inhibition of cardiac differentiation by knockdown of FN suggest that FN binding to α4β1 or αVβ1 plays a key role. Although

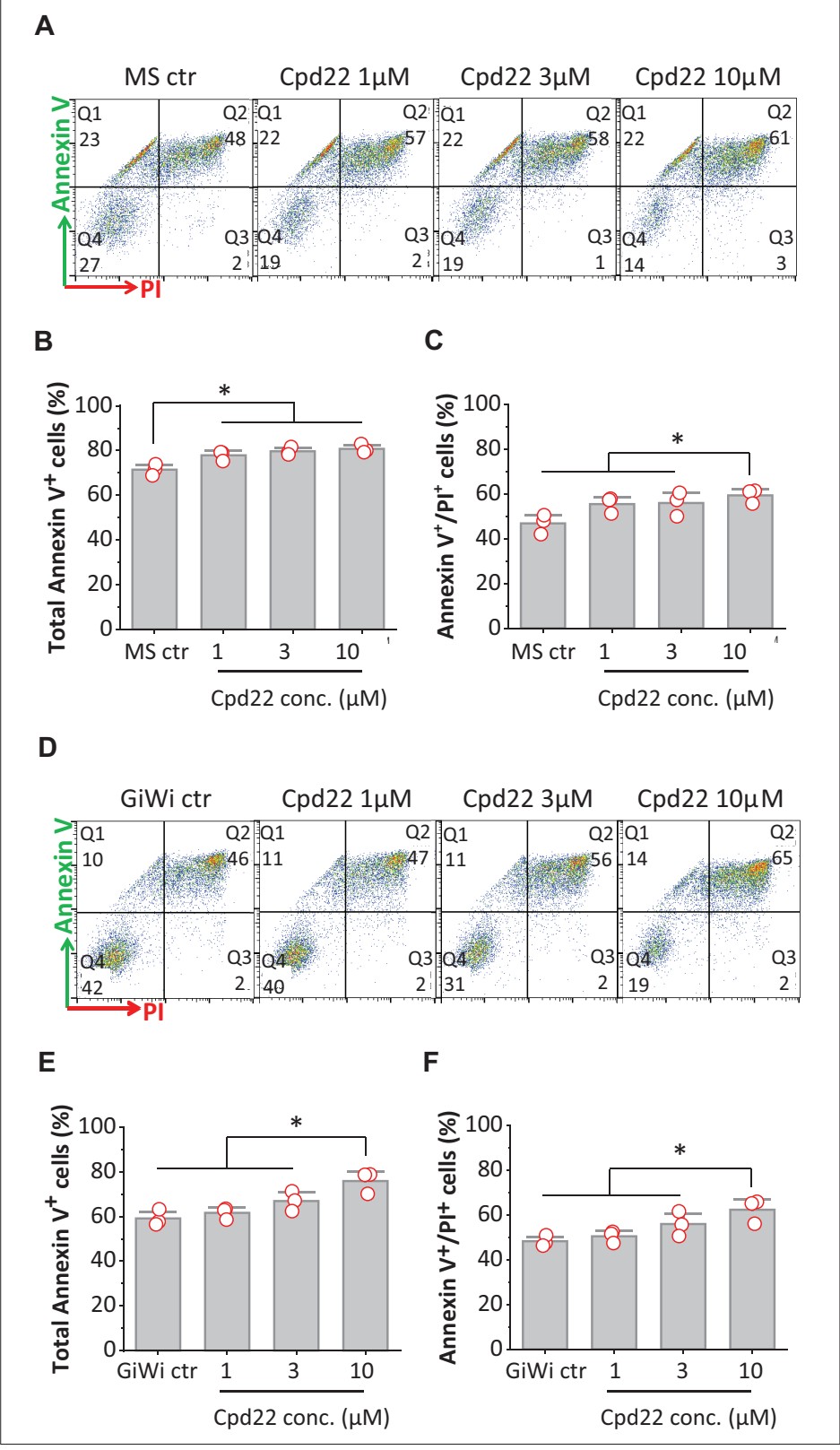

**Figure 8.** Inhibition of ILK promotes apoptosis of differentiating hPSCs at mesoderm formation in the matrix sandwich and GiWi protocols. (**A, D**) Representative flow cytometry plots of cells collected at 18 hr of differentiation, labeled by an antibody recognizing AnnexinV and stained by propidium iodide (PI) in the Matrix Sandwich (MS) protocol (**A**) and GiWi protocol (**D**) in absence or presence of the ILK inhibitor cpd22.

*Figure 8 continued on next page*

*Figure 8 continued*

(**B, C**) Average total apoptotic cells (Annexin V$^+$) and late apoptotic cells (Annexin V$^+$/PI$^+$) in the MS protocol in absence or presence of cpd22. (**E, F**) Average of total apoptotic cells of (Annexin V$^+$) and late apoptotic cells (Annexin V$^+$/PI$^+$) in the GiWi protocol in absence or presence of cpd22. N=3 biological replicates. Data are from DF19-9-11T iPSC line. Error bars represent SEM. *p<0.05, one-way ANOVA with post-hoc Bonferroni test.

The online version of this article includes the following figure supplement(s) for figure 8:

**Figure supplement 1.** Inhibition of ILK promotes apoptosis of differentiating H1 ESCs at the mesoderm formation in the Matrix Sandwich protocol.

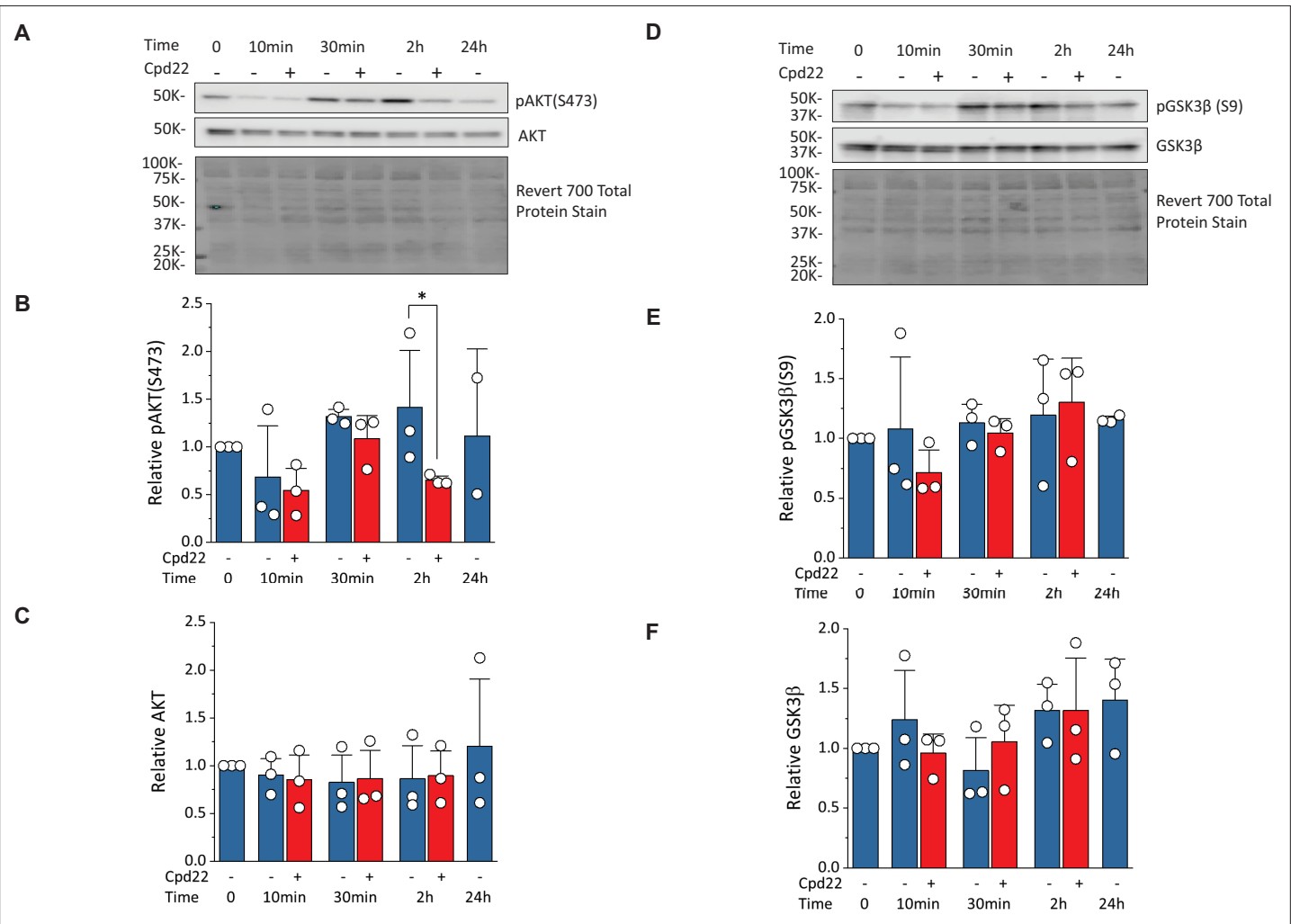

**Figure 9.** Inhibition of ILK reduces pAKT(Ser473) without changing pGSK3β(Ser9) at mesoderm formation in the matrix sandwich protocol. (**A**) Representative western blot analysis of pAKT(Ser473) and total AKT as well as total protein staining from day 0 (Time 0) to day 1 (24 hr) of the matrix sandwich protocol with or without cpd22 treatment. (**B, C**) Densitometric analysis of pAKT(Ser473) (**B**) and AKT (**C**) normalized to total protein and plotted relative to time 0 for three separate experiments. (**D**) Representative Western blot analysis of pGSK3β(Ser9) and total GSK3β as well as total protein staining from day 0 (Time 0) to day 1 (24 hr) of the matrix sandwich protocol with or without cpd22 treatment. (**E, F**) Densitometric analysis of pGSK3β(Ser9) (**E**) and total GSK3β (**F**) normalized to total protein and plotted relative to time 0 for three separate experiments. Data in (**B, C, E, F**) are presented as mean ± SEM. *p<0.05, paired T-test comparing with and without cpd22.

The online version of this article includes the following source data for figure 9:

**Source data 1.** Raw unedited gels and blots and images with the uncropped gels and blots for all three experiments performed with the relevant bands labelled.

the integrin heterodimer α5β1 is a key receptor for FN, our result shows block of integrin α5 does not inhibit cardiac differentiation, consistent with the effect of embryonic knockout of integrin α5 in mice which resulted in posterior mesoderm defects but preserved early cardiac development (*Yang et al., 1993*). Furthermore, downstream ILK-mediated signaling is implicated in the differentiation process. FN also contributes to the mechanical and structural properties of the ECM and is critical to migration of mesodermal cell populations as demonstrated in mouse and zebrafish embryos (*George et al., 1993*; *Trinh and Stainier, 2004*). Furthermore, FN can provide a reservoir for growth factors as well as serve as a coactivator with growth factors (*Hynes, 2009*). For example, FN, through its second heparin binding domain (FN III12-14), binds a variety of growth factors in the PDGF/VEGF, FGF, and TGF-β families (*Martino and Hubbell, 2010*; *Wijelath et al., 2006*). FN can also interact with binding proteins for growth factors such as IGF-binding protein-3 and latent TGF-β binding protein (*Dallas et al., 2005*; *Gui and Murphy, 2001*). FN and growth factors can be costimulatory at cellular adhesions (*Miyamoto et al., 1996*). Although the precise role of FN-associated growth factor signaling during early development is little understood, a study in *Xenopus* did demonstrate that FN-bound PDGF-AA provides guidance in mesendoderm migration during gastrulation (*Smith et al., 2009*). Future studies will be required to define the specific mechanisms by which FN contributes to cardiac-directed differentiation of hPSCs and thus enable further improvements in differentiation methods.

Overall, our study identifies several defined ECM proteins, including LN111, LN521, and FN, that can be used successfully for seeding and differentiating hPSCs to CMs. Regardless of the seeding substrate, a critical level of FN in the ECM is required for successful formation of mesodermal cells in the cardiac differentiation using the Activin A/BMP4/bFGF protocol. The FN in the ECM acts at least in part by binding integrins of α4β1 and αVβ1 leading to stimulation of ILK signaling. Ultimately, the choice of ECM substrate used for cardiac differentiation protocols will be based not only on effective outcomes but also on the cost and availability of these defined substrates. The refinement of cardiac differentiation protocols is of growing importance as applications using hPSC-CMs expand including clinical applications.

## Materials and methods

### Human PSC culture

Human ESC line H1 and iPSC line DF19-9-11T were used in this study. Human PSCs were cultured on Matrigel (GFR, BD Biosciences) coated 6-well plates in mTeSR1 medium. Cells were passaged using Versene solution (Gibco) every 4–5 days as previously described (*Zhang et al., 2012*).

### Cardiac differentiation of hPSCs

For cardiac differentiation with the matrix sandwich protocol as shown in *Figure 1A*, the tested matrix proteins were overlaid on the hPSC culture after 2 (d-2) and 4 (d0) days of seeding when the monolayer of cells reached 80% and 100% confluence. Human PSCs were dissociated with 1 ml/well Versene solution at 37 °C for 5 min and seeded on Matrigel, LN111 (BioLamina, 1 µg/cm$^2$), LN521 (BioLamina, 0.5 µg/cm$^2$) and FN (Corning 354008, 2.5 µg/cm$^2$) coated 12-well plates at the density of 6 × 10$^5$ cells/well in mTeSR1 medium supplemented with 10 µM ROCK inhibitor (Y-27632, Tocris). The medium was changed daily. The amount of matrix proteins in overlay was: 8.7 µg/cm$^2$ Matrigel; LN111, LN521, FN and COL4 was 0.3 µg/cm$^2$. For cardiac differentiation with the monolayer based protocol, the matrix overlay was not added. The cells were fed with fresh mTeSR1 medium from d-3 to d0. At d0 the medium was changed to RPMI +B27 without insulin and supplemented with growth factors. The concentration of growth factors were: Activin A (R&D) at 100 ng/ml, BMP4 (R&D) at 2.5 ng/ml, and bFGF (Waisman Biomanufacturing, UW-Madison) at 10 ng/ml. The volume of medium was 0.8 ml at d0-d1 and 1.5 ml at d1-d5. The medium was changed to RPMI +B27 with insulin, and cells were fed every other day and cultured until d15 for flow cytometry analysis.

For cardiac differentiation with the GiWi protocol, human PSCs were dissociated with 1 ml/well Versene solution at 37 °C for 5 min, and seeded on Matrigel coated 12-well plates at the density of 6–10 × 10$^5$ cells/well in mTeSR1 medium supplemented with 10 µM ROCK inhibitor (Y-27632). Cells were cultured in mTeSR1 medium with medium change daily until reached 100% confluence. At d0, the medium was changed to 1 mL RPMI +B27 without insulin and supplemented with 12 µM CHIR99021 (Tocris), and changed to 2 mL RPMI +B27 without insulin after 24 hr. Seventy-two hr after

addition of CHIR99021, a combined medium was prepared by collecting 1 mL of medium from the wells and mixing with same volume of fresh RPMI +B27 without insulin medium and supplemented with 5 µM IWP2 (Tocris). The medium was changed to 2 mL RPMI +B27 without insulin on d5, and to RPMI +B27 with insulin starting from d7. Cells were fed every other day and cultured until d15 for flow cytometry analysis.

## Generation of inducible *FN1* knockdown hPSC clones
Human PSCs maintained in mTeSR1 medium were used for lentivirus transfection. The cloning strategy is shown schematically in *Figure 4—figure supplement 4*.

### Production of lentivirus particles
HEK 293 TN cells (SBI) were used for lentivirus production. Briefly, $4.5 \times 10^6$ cells were plated in a 10 cm dish on the day before transfections. Lipofectamine 2000 (Invitrogen) was used for transfections (1:2 ratio). Transfection conditions used were – 7 µg lentivirus plasmid, 10 µg psPAX2 (Addgene plasmid #12260 - packaging), 5 µg pMD2.G (Addgene plasmid #12259 - envelope). Transfection media was incubated for 15–16 hr, after which it was replaced with 5 mls of mTeSR1medium. Lentivirus supernatant was collected after 48–52 hr, filtered (0.45 µM – Millipore) and frozen down at –80 °C. Lentivirus supernatants were thawed in 37 °C water bath immediately before infection.

### hPSCs transfection and neomycin selection
Human PSCs at the 80% confluency after split were incubated with the lentivirus in mTeSR1 medium in the presence of 8 µg/ml Polybrene (Sigma) for 24–42 hr. After lentivirus transfection, hPSCs were washed with PBS and recovered in mTeSR1 medium for 2–3 days with medium change daily, cells were split once during this recovery. The concentration of G418 (Life technologies) for neo-resistant selection for hPSC lines was determined by the kill curve (*Figure 4—figure supplement 5*). G418 (100 mg/ml) was diluted in mTeSR1 medium with the final concentrations of 100 µg/ml for H1 ES cells and 75 µg/ml for DF19-9-11T, and hPSCs were under neomycin selectin up to 11 days before single cell isolation and cloning.

### Single-cell isolation and cloning
Human PSCs resistant to neomycin were incubated with Accutase (Gibco) at 37 °C for 5–10 min, and resuspended in mTeSR1 medium to microscopically examine for single cells. If cells were still clustered, cells were spun down and washed with PBS, and underwent Accutase treatment for another time period as above. Single cells were plated in 6-well Matrigel coated plates at the low density of 50–100 cells/cm² with 10 µM Rock inhibitor (Y-27632). Single cells were grown in mTeSR1 medium for 4–7 days, clones were picked using a P200 pipette tips and transferred to microtubes with 30 µl Versene solution (pre-warmed to room temperature), and incubated at 37 °C for 5 min. Cells were pipette up and down several times and equally dispersed in two wells of Matrigel coated 24-well plates in mTeSR1 medium.

### Doxycycline induction
To induce *FN1* shRNA and mCherry expression, doxycycline (Sigma, 2–8 µg/ml) in mTeSR1 medium was added to the hPSC clones cultured in the 12-well plate for 2–3 days. mCherry expression was examined with EVOS microscope (Life Technologies); *FN1* expression were examined by quantitative RT-PCR.

### Expansion and cryopreservation of clones
Selected *FN1* knockdown positive clones were expanded in mTeSR1 medium in 6-well culture. Cells were cryopreserved in 90% FBS, 10% DMSO and 10 µM ROCK inhibitor in liquid nitrogen (LABS20K, Taylor-Wharton).

## Immunocytochemistry
For imaging with an EVOS microscope (Life Technologies), hPSCs were cultured either in 6-well plates, or seeded and differentiated in 12-well plates, coated with Matrigel or the ECM proteins. For

imaging with confocal microscope (Leica SP5II), cells were seeded or plated on glass coverslips coated with Matrigel or the ECM proteins. Cells were fixed with 4% paraformaldehyde for 15 min at room temperature for labeling with different markers. For intracellular markers, cells were permeabilized in 0.2% Triton X-100 (Sigma) for 1 hr at room temperature. For labeling the ECM proteins, cells were not permeabilized. After fixation and permeablization, samples were blocked with 5% non-fat dry milk (Bio-Rad) in 0.2% Triton X-100 solution and incubated for 2 hr at room temperature on a rotator followed by two washes with PBS. Primary antibodies (*Supplementary file 1a*) were added in 1% BSA in PBS solution with or without 0.1% Triton X-100 depending on the markers to label, and incubated overnight at 4 °C. Samples were washed with 0.2% Tween 20 in PBS twice and 1 X PBS twice. Secondary antibodies specific to the primary IgG isotype were diluted (1:1000) in the same solution as the primary antibodies and incubated at room temperature for 1.5 hours in dark on a rotator. Samples were washed with 0.2% Tween 20 in PBS twice and 1 X PBS twice. Nuclei were labeled with Hoechst or DAPI. Confocal images were analyzed with the Leica LAS AF Lite software.

## Flow cytometry

Cells were detached from cell culture plates by incubation with 0.25% trypsin-EDTA (Invitrogen) plus 2% chick serum (Sigma) for 5 min at 37 °C. Cells were vortexed to disrupt the aggregates followed by neutralization by adding an equal volume of EB20 medium (*Zhang et al., 2009*). Cells were fixed in 1% paraformaldehyde in a 37 °C water bath for 10 min in the dark, permeabilized with ice-cold 90% methanol for 30 min on ice. Cells were washed once in FACS buffer (PBS without Ca/Mg$^{2+}$, 0.5% BSA, 0.1% NaN$_3$) plus 0.1% Triton, centrifuged, and the supernatant was discarded leaving about 50 μl. See *Supplementary file 1a* for the primary antibodies information. For labeling cTnT, the primary antibody was diluted in 50 μl FACS buffer plus 0.1% Triton and added to each sample for a total sample volume of 100 μl. Samples were incubated with the primary antibodies overnight at 4 °C. For co-labeling of Brachyury and Sox17, the cells were incubated with the conjugated antibodies for half an hour at room temperature in dark. Cells were washed once with 3 ml FACS buffer plus 0.1% Triton and resuspended in 300–500 μl FACS buffer plus Triton for analysis. Please refer to Supplementary Material for details of the primary antibodies. For the secondary antibody labeling for cTnT, cells were washed once with 3 ml FACS buffer plus 0.1% Triton after the primary antibody labeling, centrifuged, and supernatant discarded leaving ~50 μl. Secondary antibody specific for the primary IgG isotype was diluted in FACS buffer plus Triton in a final sample volume of 100 μl at 1:1000 dilution. Samples were incubated for 30 min in the dark at room temperature, washed in FACS buffer plus Triton and resuspended in 300–500 μl FACS buffer plus Triton for analysis. Data were collected on a FACSCaliber flow cytometer (Beckton Dickinson) and analyzed using FlowJo.

## Quantification of FN immunofluorescence by ImageJ analysis

The FN immuno-fluorescence images of d-3, d-2, d-1, d0 matrix sandwich culture, and d0 monolayer culture were evenly split into 4 squares with online image splitting tool (https://www.imgonline.com.ua/eng/cut-photo-into-pieces.php), and imported to ImageJ Fiji software. The control images without primary antibody labeling were also imported and analyzed in the same way by ImageJ Fiji software. The gray value was used to represent the intensity of fluorescence, ranging from 0 (black, low intensity) to 256 (white, high intensity). For mean and standard deviation of gray values, the entire area of each image was selected, and the values were retrieved by selecting Menu bar >Analyze > Measure. The mean gray value was generated from ImageJ by summing each pixel's gray value divided by the number of pixels in the selected area. The final mean fluorescence value was obtained by subtracting the mean gray value of the control image from FN immuno-labeled images. The mean of fluorescence intensity of each quartiles from each biological replicate in each group were represented in the box plots using Origin v9.

## RT-PCR and quantitative RT-PCR

Cell samples were collected using 0.25% trypsin-EDTA (Invitrogen) to remove the cells from cell culture plates. Total RNA was purified using QIAGEN RNeasy Mini kit. Possible genomic DNA contamination was removed by RNase-Free DNase Set (QIAGEN) with the RNeasy columns, or by DNase I (Invitrogen) treatment for 15 min at room temperature. A total of 500 ng of total RNA was used for Oligo(dT)$_{20}$ – primed reverse transcription using SuperScript III First-Strand Synthesis System

(Invitrogen). Quantitative RT-PCR was performed using Taqman PCR Master Mix and Gene Expression Assays (Applied Biosystems, *Supplementary file 1b*) in triplicate for each sample and each gene. A total of 0.5 µl of cDNA from RT reaction was added as template for each Q-PCR reaction. The expression of genes of interest was normalized to that of *GAPDH*. RT-PCR was carried out using Platinum Taq DNA Polymerase (Invitrogen) or Gotaq Master Mix (Promega) and then subjected to 2% agarose gel electrophoresis. PCR conditions included denaturation at 94 °C for 30 s, annealing at 60 °C for 30 s, and extension at 72 °C for 1 min, for 35 cycles, with 72 °C extension for 7 min at the end. *ACTB* (β-actin) was used as an endogenous control.

## Antibody blocking and small molecule inhibition experiments

Monoclonal antibodies in supernatant without sodium azide were used for the blocking experiments. Blocking antibodies, P5D2 (anti- human integrin β1), P1D6 (anti-human integrin α5), P3G8 (anti-human integrin αV) and P4G9 (anti-human integrin α4) (all from Developmental Studies Hybridoma Bank), and ILK inhibitor cpd22 (EMD Millipore) were diluted in the media to make the final concentrations and added to the culture at the indicated time points.

## Immunoblotting

Cell pellets were re-suspended in modified RIPA buffer (50 µl per $10^6$ cells) containing 50 mM Tris-HCl pH 8, 150 mM NaCl, 1% Triton X-100, 1 mM EDTA pH 8, 1 mM PMSF, and 1 X Protease/Phosphatase Inhibitor Cocktail (Cell Signaling Technology, cat# 5872) by pipetting solution up and down several times and incubated 30 min at 4 °C mixing by end-over-end rotation. Lysates were sonicated using a Fisherbrand Model 120 Sonic Dismembrator (4×5 s bursts at 30% amplitude with 10 s between bursts) and centrifuged (16,100×*g*, 20 min, 4 °C) to clarify. The supernatants were recovered and protein content was determined using Bio-Rad Protein Assay Dye Reagent Concentrate (cat# 5000006) following the manufacturer's instructions.

To assess changes in AKT and GSK3β phosphorylation, as well as changes in the expression of ILK, AKT, and GSK3β, 25 µg of cell lysate from each condition was resolved by SDS-PAGE. Proteins were transferred to Amersham Hybond P Western blotting membrane (0.2 µm pore size, PVDF, Sigma-Aldrich, cat# GE10600021) at 10 V overnight in a cold room (4 °C). After transfer, total protein loading was assessed by staining with Revert 700 Total Protein Stain (LI-COR, cat# 926–11010) in adherence with the manufacturer's instructions. To block, membranes were incubated in TBST with 10% (w/v) nonfat dry milk for 1 hr at room temperature followed by incubation in primary antibody solution containing either 5% (w/v) BSA (AKT-Ser473 and GSK3β-Ser9) or nonfat dry milk (all other antibodies) and diluted primary antibodies in TBST overnight at 4 °C (see *Supplementary file 1c* for primary antibody information). The membranes were removed from primary antibody solution and washed 5×5 min with TBST followed by incubation in TBST with 3% (w/v) nonfat dry milk and 1:1,000 diluted secondary antibodies for 1 hr at room temperature. Subsequently, membranes were washed 5×5 min with TBST, overlayed with Amersham ECL Prime Western Blotting Detection Reagent (Sigma-Aldrich, cat# GERPN2236), and imaged using an Odyssey XF Imaging System (LI-COR).

Membranes were probed first with phospho-specific antibodies, stripped using Restore Western Blot Stripping Buffer (Thermo Scientific, cat# 21059) in accordance with the manufacturer's instructions, and reprobed with pan-protein antibodies. Band densities were quantified using Image Studio Lite Quantification Software (Ver 5.2, LI-COR). The phospho-specific and pan-protein signals were normalized to total protein. Data are from three biological replicates with one or more technical replicates, and are presented as mean ± SEM.

## Statistics

Data are presented as mean ± SEM. Technical and biological replicates are as indicated for each dataset. For datasets with normal distributions, statistical significance was determined by Student's t-test for two groups or one-way ANOVA for multiple groups with post-hoc test using Bonferroni

method. Statistical analysis was performed using Origin, v9, p<0.05 was considered statistically significant.

## Acknowledgements

The authors thank Ms. Yukun Li for assistance with the image analysis using ImageJ and UW-Madison Carbone Cancer Center Flow Lab for providing the flow cytometry facility. The work was funded by NIH R01 HL129798 (TJK), NIH 1R01HL46652 (TJK), NIH 1R01HL148059 (SPP), NIH U01HL134764 (TJK), NIH R01EB007534 (SPP), and NSF EEC-1648035 (SPP and TJK).

## Additional information

### Competing interests

Timothy J Kamp: is a consultant for Fujifilm Cellular Dynamics Incorporated, a stem cell company. The other authors declare that no competing interests exist.

### Funding

| Funder | Grant reference number | Author |
| --- | --- | --- |
| National Institutes of Health | 1U01HL134764 | Timothy J Kamp |
| National Institutes of Health | 1R01HL148059 | Sean P Palecek |
| National Science Foundation | 1648035 | Sean P Palecek Timothy J Kamp |
| National Institutes of Health | 1R01HL46652 | Timothy J Kamp |
| National Institutes of Health | 1R01EB007534 | Sean P Palecek |
| National Institutes of Health | 1R01H129798 | Timothy J Kamp |

The funders had no role in study design, data collection and interpretation, or the decision to submit the work for publication.

### Author contributions

Jianhua Zhang, Conceptualization, Formal analysis, Investigation, Methodology, Validation, Writing – original draft, Writing – review and editing; Zachery R Gregorich, Formal analysis, Investigation, Methodology, Writing – review and editing; Ran Tao, Gina C Kim, Juliana L Carvalho, Investigation, Methodology; Pratik A Lalit, Methodology, Resources; Yogananda Markandeya, Formal analysis, Investigation, Methodology; Deane F Mosher, Conceptualization, Writing – review and editing; Sean P Palecek, Conceptualization, Funding acquisition, Supervision, Writing – review and editing; Timothy J Kamp, Conceptualization, Funding acquisition, Investigation, Project administration, Supervision, Writing – original draft, Writing – review and editing

### Author ORCIDs

Jianhua Zhang http://orcid.org/0000-0001-5134-7918
Juliana L Carvalho http://orcid.org/0000-0002-1423-0523
Yogananda Markandeya http://orcid.org/0000-0001-9783-2046
Timothy J Kamp http://orcid.org/0000-0003-2103-7876

### Decision letter and Author response

Decision letter https://doi.org/10.7554/eLife.69028.sa1
Author response https://doi.org/10.7554/eLife.69028.sa2

## Additional files

### Supplementary files
• Supplementary file 1. Tables of primary antibodies and qRT-PCR primers. (a) Primary antibodies used in immunocytochemistry (ICC) and flow cytometry (FC). (b) Primers for quantitative RT-PCR. (c) Primary antibodies used for immunoblotting.

• Transparent reporting form

### Data availability
Raw data are uploaded as source files with the submission.

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
