## [Editor Report]

We found this study important for advancing derivation of cardiac cells from human pluripotent stem cells, as it convincingly supports the critical role of fibronectin in the formation of precardiac mesoderm. We believe that the work will be of interest to developmental biologists, stem cell biologists, and engineers as they work to optimize substrates used for preparation of cardiomyocytes and supporting cardiac cells.

---

## [Decision Letter]

**Decision letter after peer review:**

Thank you for submitting your article "Cardiac Differentiation of Human Pluripotent Stem Cells Using Defined Extracellular Matrix Proteins Reveals Essential Role of Fibronectin" for consideration by *eLife*. Your article has been reviewed by 2 peer reviewers, and the evaluation has been overseen by a Reviewing Editor and Didier Stainier as the Senior Editor. The following individual involved in review of your submission has agreed to reveal their identity: Milica Radisic (Reviewer #1).

Essential revisions:

1. Many of the studies were conducted using a single hiPSC line. Some studies were conducted with an hESC line, but not the most critical experiments including demonstration of a lack of FN at early time points when cultured on LN-111. Using both cell lines in these critical experiments is recommended.

2. In addition to direct signaling through cell surface receptors, in vivo ECM can sequester and then slowly release growth factors. Is it possible that Fb is particularly suitable for sequestering and release of these factors in the context of early mesoderm induction and cardiac differentiation?

3. Given that cardiac differentiation cultures are cell density sensitive, is it possible that blocking of integrin beta1/ILK can result in lower cell viability that translates into lower cell density, which in turn affects the cardiac differentiation?

4. Β 1 integrin subunit is necessary for engagement of most ECM proteins and therefore downstream outcomes cannot be linked directly to FN. An opportunity is missed to identify the heterotrimers associated with the observed differentiation outcomes.

5. The studies of cpd22 were conducted without small-molecule control and knockdown to avoid off-target effects of the small molecule, and there is no indication of whether the effect is linked to GSK3b, PI3K, or both. With the identification of the critical integrin heterodimer(s), it would be possible to block these and look at downstream phosphorylation of ILK and other downstream signaling molecules.

*Reviewer #2 (Recommendations for the authors):*

Additional Major Criticisms for the Author:

1. Past research is cited in the introduction with respect to previous studies ECM but the choice of FN, laminin 111, and laminin 521 are not clearly explained.

2. qRT-PCR results in Figure 3 would be strengthened by the inclusion of other conditions in the analysis such as FN as positive control and Laminin 521 as negative in comparison to the Laminin 111 condition.

---

## [Author Response]

Essential revisions:1. Many of the studies were conducted using a single hiPSC line. Some studies were conducted with an hESC line, but not the most critical experiments including demonstration of a lack of FN at early time points when cultured on LN-111. Using both cell lines in these critical experiments is recommended.

We agree with the reviewer that confirming key findings in multiple cell lines is critical to demonstrate robust findings. We tested hiPSC line DF19-9-11T and hESC line H1 for almost every experiment, but given space constraints much of the confirmatory data were presented as supplemental figures. We apologize that the text did not always clearly state when multiple cell lines were used, but we have revised where necessary to indicate this. In addition, we have updated some figure legends to clearly state which cell lines were used. In addition, we have provided additional confirmatory data for our integrin antibody blocking experiments in both lines. Here is a summary of the experiments presenting data for experiments done in parallel in 19-9-11 and H1 and our revisions to refer to both sets of data more clearly:

a) Test of defined ECM proteins (LN111, LN521 and FN) in Figure 1B, C for 19-9-11 and Figure 1 - figure supplement 1, 3 for H1. Line 134-135 revised to state, “…we tested overlay of defined ECM proteins in the matrix sandwich protocol using DF19-9-11T iPSCs and H1 ESCs.”

b) Endogenous FN production on LN111 for DF19-9-11T iPSCs (Figure 2C) and H1 ESCs (Figure 2—figure supplement 1 ). Line 168-171, “…no detectable FN ECM at day -3 and -2, similar to the Matrigel/Matrigel sandwich culture; however, by day 0, dense fibrillary FN ECM was present in the cell culture on LN111 coated surface (Figure 2C, DF19-9-11T iPSCs; Figure 2 —figure supplement 1, H1 ESCs).” The time course is more abbreviated in Figure 2 —figure supplement 1 as our experience showed minimal detectable FN under any conditions on day -3 and -2. Therefore we focused on day -1 and day 0 for the supplemental experiments in H1 which confirmed the increase in FN from day -1 to day 0.

c) FN1 shRNA knockdown clones were generated in both H1 ESCs and DF19-9-11T iPSCs (Figure 4 —figure supplement 2, 4).

d) FN knockdown studies were conducted in multiple clones from both 19-9-11 and H1 transgenic lines. FN knockdown and exogenous FN rescue experiments were carried out in multiple clones of H1 as shown in Figure. 4 and Figure 4 —figure supplement 3. With the evidence of inhibition of cardiac differentiation by knockdown of FN, gene expression and flow cytometry for Brachyury^+^ cells were examined in the same H1 knockdown clones (Figure 5). Furthermore, flow cytometry evaluation of Brachyury^+^ cells was tested in the DF19-9-11T FN knockdown clones as well (Figure 5 —figure supplement 1, 2).

e) The integrin antibody blocking for β1, α5, αV (added in revision), and α4 (added in revision) were tested in both 19-9-11 (Figure 6) and H1 (Figure 6 —figure supplement 1). Lines 304-318 “Adding P5D2 at day -2 did not block cardiac differentiation; however, adding P5D2 at day 0 significantly inhibited cardiac differentiation as measured by flow cytometry of the cTnT^+^ cells using DF19-9-11T iPSCs and H1 ESCs (Figure 6B, Figure 6 —figure supplement 1A)… and block of integrin α4 showed significant inhibition of hPSC cardiac differentiation (Figure 6D, Figure 6 —figure supplement 1C).”

f) The ILK inhibitor, cpd22, was tested using 19-9-11 line in Matrix Sandwich protocol (Figure 7A, B) and GiWi protocol (Figure 7C, D), and using H1 line in MS protocol (Figure 7 —figure supplement 1).

g) Newly added in the revision, the effect of ILK inhibition by cpd22 on apoptosis was examined in both 19-9-11 (Figure 8A-C) and H1 (Figure 8 —figure supplement 1) in the Matrix Sandwich protocol, as well as in the GiWi protocol using 19-9-11 (Figure 8D-F).

2. In addition to direct signaling through cell surface receptors, in vivo ECM can sequester and then slowly release growth factors. Is it possible that Fb is particularly suitable for sequestering and release of these factors in the context of early mesoderm induction and cardiac differentiation?

Thank you for this important comment. Yes, it is possible and likely that fibronectin impacts growth factor signaling by sequestering and releasing these factors. We now specifically address this question in the Discussion highlighting relevant literature. Lines 470-479 state, “Furthermore, FN can provide a reservoir for growth factors as well as serve as a coactivator with growth factors (Hynes, 2009). For example, FN through its second heparin binding domain (FN III12-14) binds a variety of growth factors in the PDGF/VEGF, FGF, and TGF-β families (Martino and Hubbell, 2010; Wijelath et al., 2006). FN can also interact with binding proteins for growth factors including IGF-binding protein-3 and latent TGF-β binding protein (Dallas et al., 2005; Gui and Murphy, 2001). FN and growth factors can be costimulatory at cellular adhesions (Miyamoto et al., 1996). Although the precise role of FN-associated growth factor signaling during early development is little understood, a study in *Xenopus* did demonstrate that FN bound PDGF-AA provides guidance in mesenoderm migration during gastrulation (Smith et al., 2009).” This discussion highlights the multiple possible interacting signaling molecules with FN that potentially contribut to the cardiac differentiation process, but mechanistically dissecting out the role of these interactions is beyond the scope of this manuscript.

3. Given that cardiac differentiation cultures are cell density sensitive, is it possible that blocking of integrin beta1/ILK can result in lower cell viability that translates into lower cell density, which in turn affects the cardiac differentiation?

We agree with the reviewer that the monolayer hPSC-CM differentiation protocols are cell density sensitive. Cell density is typically optimized for the confluence of hPSCs at the initiation of differentiation (day 0 in our manuscript) which was the starting point for testing integrin subunit and ILK block. How cell density impacts differentiation after day 0 as the cells begin to undergo the epithelial-to-mesenchymal transition is not clear to us from the existing literature as this is difficult to experimentally manipulate independent of the cell density at day 0. There is a dramatic change in cell viability and density during this early time window (see Figure 5B). Nevertheless, we agree that a decrease in cell viability or increased apoptosis in the presence of the blocking interventions may be a critical mechanistic feature. For that reason, we performed additional experiments examining cell survival using annexin V labeling to identify apoptotic cells and π for cell viability in the presence and absence of cpd22 (see new Figure 8). The results show that cpd22 addition on day 0 induces a concentration-dependent increase in apoptosis after 18 hr on top of the high background level of apoptosis in both the matrix sandwich and GiWi protocols. Although the increase in apoptosis by cdp22 correlates with the inhibition of cardiac differentation in this time window by cpd22, it does not clarify whether the impact on differentation is due to a reduction in density of the surviving cells or apoptosis of the essential population of cells forming mesoderm. Regardless, the new data confirm the reviewer’s suspicion that changes cell density after day 0 can help explain the impact of blocking integrin β1 and ILK signaling. Lines 358-362 introduce these new data,

“Because inhibition of ILK by cpd22 significantly inhibited cardiac differentiation in the Matrix Sandwich protocol, we next investigated apoptosis after 18 hours of cpd22 addition on day 0 in the Matrix Sandwich protocol. To identify apoptotic and necrotic cells, annexin V/ propidium iodide (PI) double staining was used…”

4. Β 1 integrin subunit is necessary for engagement of most ECM proteins and therefore downstream outcomes cannot be linked directly to FN. An opportunity is missed to identify the heterotrimers associated with the observed differentiation outcomes.

We greatly appreciate this comment and have done additional experiments to address it. In the revised manuscript, we test a series of blocking antibodies to integrin α subunits in addition to α5 that are known to bind FN as a heterodimer with integrin β1. In the revised manuscript, we now state (lines 294-300),

“Of the 24 known heterodimeric integrin receptors, 13 have been shown to bind FN (Bachmann et al., 2019; Bharadwaj et al., 2017; Hynes, 2002; Ruoslahti, 1991; Wu et al., 1995). Review of RNA-seq data from undifferentiated iPSCs shows expression of integrin subunits associated with FN binding including integrin α3, α4, α5, αV, β1, β5, and β8 (Zhang et al., 2019). Of these integrin subunits, knockout studies have implicated only α4, α5, αV, and β1with various developmental defects and β1with various developmental defects impacting the heart (Hynes, 2002), so we focused our studies on these integrins…”

In short, we found that blocking integrin α4 or αV inhibits cardiac differentiation (revised Figure 6D, Figure 6 – figure supplement 1). Thus we conclude on lines 326 and 327,

“…integrin α4β1 and αVβ1 heterodimers are likely key mediators of the FN effect on early differentation stages.”

5. The studies of cpd22 were conducted without small-molecule control and knockdown to avoid off-target effects of the small molecule, and there is no indication of whether the effect is linked to GSK3b, PI3K, or both. With the identification of the critical integrin heterodimer(s), it would be possible to block these and look at downstream phosphorylation of ILK and other downstream signaling molecules.

We appreciate the reviewers’ important mechanistic questions. Indeed many questions remain regarding downstream signaling. Because there are no published and available chemical analogues to use as a small molecular control, we were unable to perform this experiment. We also were unable to generate an effective ILK inducible knockdown PSC line. However, we did focus our attention on signaling pathways known to be downstream of ILK including GSK3β and AKT looking at both total protein and phosphorylation of these kinases at residues correlated with activity pGSK3β (Ser9) and pAKT (Ser473). Our results show that treatment of Day 0 differentiating cells with cpd22 lead to a statistically significant reduction in pAKT at 2 hr relative to control, but there were no significant changes in pGSK3β induced by cpd22 treatment (new Figure 9). The reduction in pAKT correlates with an increase in apoptosis observed in response to cpd22 treatment (Figure 8). Although we did make progress identifying α4β1 and αVβ1 as the likely critical integrin heterodimers, we were unable to perform blocking antibody experiment time courses with the series of blocking integrin antibodies due to limitations in available reagents and timely revision during the pandemic.

Reviewer #2 (Recommendations for the authors):Additional Major Criticisms for the Author:1. Past research is cited in the introduction with respect to previous studies ECM but the choice of FN, laminin 111, and laminin 521 are not clearly explained.

We have revised the Introduction to clarify the rationale for our choices of defined ECM to test. Lines 107-111 state:

“We chose to test LN111 (the dominant laminin isoform in Matrigel), LN521 (demonstrated adherence and culture of hPSCs [8]), FN (implicated in embryonic developmental studies [12-14]) and collagen (common ECM protein used for in vitro cell adherence).”

2. qRT-PCR results in Figure 3 would be strengthened by the inclusion of other conditions in the analysis such as FN as positive control and Laminin 521 as negative in comparison to the Laminin 111 condition.

Thank you for this suggestion. We understand the reviewers goal for positive and negative controls, but our prior publication provides a matrix sandwich control pattern of comparable gene expression in Zhang et al., 2012, Circ Res 111(9):1125-36 in Figure 2 and online Figure II. We did not do the time course of gene expression for Ln 521 overlay as we focused on the effective differentation conditions. In retrospect, this may have been informative of any key differences in gene expression but these data are not available.

References

Bachmann, M., Kukkurainen, S., Hytonen, V.P., and Wehrle-Haller, B. (2019). Cell Adhesion by Integrins. Physiol Rev *99*, 1655-1699.

Bharadwaj, M., Strohmeyer, N., Colo, G.P., Helenius, J., Beerenwinkel, N., Schiller, H.B., Fassler, R., and Muller, D.J. (2017). alphaV-class integrins exert dual roles on alpha5beta1 integrins to strengthen adhesion to fibronectin. Nat Commun *8*, 14348.

Dallas, S.L., Sivakumar, P., Jones, C.J., Chen, Q., Peters, D.M., Mosher, D.F., Humphries, M.J., and Kielty, C.M. (2005). Fibronectin regulates latent transforming growth factor-β (TGF β) by controlling matrix assembly of latent TGF β-binding protein-1. J Biol Chem *280*, 18871-18880.

Gui, Y., and Murphy, L.J. (2001). Insulin-like growth factor (IGF)-binding protein-3 (IGFBP-3) binds to fibronectin (FN): demonstration of IGF-I/IGFBP-3/fn ternary complexes in human plasma. J Clin Endocrinol Metab *86*, 2104-2110.

Hynes, R.O. (2002). Integrins: bidirectional, allosteric signaling machines. Cell *110*, 673-687.

Hynes, R.O. (2009). The extracellular matrix: not just pretty fibrils. Science *326*, 1216-1219.

Martino, M.M., and Hubbell, J.A. (2010). The 12th-14th type III repeats of fibronectin function as a highly promiscuous growth factor-binding domain. Faseb J *24*, 4711-4721.

Miyamoto, S., Teramoto, H., Gutkind, J.S., and Yamada, K.M. (1996). Integrins can collaborate with growth factors for phosphorylation of receptor tyrosine kinases and MAP kinase activation: roles of integrin aggregation and occupancy of receptors. J Cell Biol *135*, 1633-1642.

Ruoslahti, E. (1991). Integrins. J Clin Invest *87*, 1-5.

Smith, E.M., Mitsi, M., Nugent, M.A., and Symes, K. (2009). PDGF-A interactions with fibronectin reveal a critical role for heparan sulfate in directed cell migration during *Xenopus* gastrulation. Proc Natl Acad Sci U S A *106*, 21683-21688.

Wijelath, E.S., Rahman, S., Namekata, M., Murray, J., Nishimura, T., Mostafavi-Pour, Z., Patel, Y., Suda, Y., Humphries, M.J., and Sobel, M. (2006). Heparin-II domain of fibronectin is a vascular endothelial growth factor-binding domain: enhancement of VEGF biological activity by a singular growth factor/matrix protein synergism. Circ Res *99*, 853-860.

Wu, C., Fields, A.J., Kapteijn, B.A., and McDonald, J.A. (1995). The role of α 4 β 1 integrin in cell motility and fibronectin matrix assembly. J Cell Sci *108 ( Pt 2)*, 821-829.

Zhang, J., Tao, R., Campbell, K.F., Carvalho, J.L., Ruiz, E.C., Kim, G.C., Schmuck, E.G., Raval, A.N., da Rocha, A.M., Herron, T.J.*, et al.* (2019). Functional cardiac fibroblasts derived from human pluripotent stem cells via second heart field progenitors. Nature communications *10*, 2238.